

# A method to retrieve mixed phase cloud vertical structure from airborne lidar

Ewan Crosbie[1,2], Johnathan W. Hair[1], Amin R. Nehrir[1], Richard A. Ferrare[1], Chris Hostetler[1], Taylor Shingler[1], David Harper[1], Marta Fenn[1,3], James Collins[1,3], Rory Barton-Grimley[1], Brian Collister[1], K. Lee Thornhill[1,2], Christiane Voigt[4,5], Simon Kirschler[4,5], Armin Sorooshian[6,7]

[1]NASA Langley Research Center, Hampton, VA 23666, U.S.A.
[2]Analytical Mechanics Associates, Inc., Hampton, VA 23666, U.S.A.
[3]Coherent Application, Inc. – Psionic LLC, Hampton, VA 23666, USA.
[4]Institut für Physik der Atmosphäre, Deutsches Zentrum für Luft- und Raumfahrt (DLR), Oberpfaffenhofen, Germany
[5]Institut für Physik der Atmosphäre, Johannes Gutenberg-Universität, Mainz, Germany
[6] Department of Chemical and Environmental Engineering, University of Arizona, Tucson, AZ 85721, U.S.A.
[7] Department of Hydrology and Atmospheric Sciences, University of Arizona, Tucson, AZ 85721, U.S.A.

*Correspondence to*: Ewan Crosbie (ewan.c.crosbie@nasa.gov)

**Abstract.** A technique was developed to provide cloud phase information using data collected by the NASA Langley airborne High Spectral Resolution Lidar systems with a particular emphasis on mixed phase cloud conditions, where boundaries and gradients in the distribution of ice and liquid water are critically important for microphysical and radiative processes. The method is based on the established use of depolarization to identify ice particles but incorporates a new method to separate the ice depolarization from the depolarization produced by multiple scattering in dense liquid clouds. Clouds assured to be liquid-only based on ambient temperature were used to train an empirical model of the multiple scattering depolarization that results at different ranges from the lidar. The method classifies lidar observations as liquid dominant, mixed phase and ice dominant and has an additional categorization for oriented ice. For evaluation of the retrieval, a two aircraft approach was used with the lidar observing the same clouds that were concurrently sampled with in situ microphysical probes. Aircraft matchups were able to track the individual cloud elements and capture marked changes in the distribution of liquid and ice across flight segments of typically 20-100 km. Qualitative features relating to localized changes in the cloud top temperature, cloud morphology and convective circulations were generally replicated between the lidar phase classification and the in situ microphysical data. Quantitative evaluation of the phase classification was carried out using a subset of fifteen cloud scenes that satisfied strict aircraft colocation and microphysical requirements. Using the in situ microphysical data, it was found that ice extinction fractions of 14% and 76% most closely matched the upper and lower bounds of the lidar mixed phase classification.



## 1 Introduction

Mixed phase clouds are an important component of the Earth's climate system (Tan et al., 2016; McCoy et al., 2015; Hofer et al., 2024; Bodas-Salcedo et al., 2019) requiring robust observational strategies and appropriate physical treatment in models. The specific arrangement of ice and liquid water within a mixed-phase cloud system is a response to the complex array of
interconnected microphysical, thermodynamic, and dynamic processes (Morrison et al., 2012; Korolev and Field, 2008). Ice particles, liquid water droplets, and mixtures thereof can occupy distinct spatial subregions of varying scales within a broader cloud system (Korolev and Milbrandt, 2022; Hogan et al., 2003; Korolev et al., 2003; Chylek and Borel, 2004; Ruis-Donoso et al., 2020; Coopman and Tan, 2023; McFarquhar et al., 2007; Kirschler et al., 2023). In supercooled polar stratus clouds, ice formation and subsequent precipitation is ubiquitous (Shupe et al., 2006; McFarquhar et al., 2007; Moser et al., 2023), but
questions remain as to the significance of ice on the longevity and climatic relevance of supercooled water layers (e.g., Silber et al., 2021).

At the microphysical scale, the formation, maintenance, and dissipation of mixed phase cloud states rely on the vapor pressure relationship between ice and water, the relative surface areas available for condensation/deposition and evaporation/sublimation, the abundance of cloud condensation nuclei to form droplets, and the primary and secondary
mechanisms available to form ice crystals (Fridlind and Ackerman, 2018; Solomon et al., 2018; Pinsky et al., 2015; Field at al., 2017, Korolev and Leisner, 2020). At larger length scales associated with dominant cloud circulations and beyond, inhomogeneities in the distribution of ice and water affect the net microphysical rates (e.g., Abel et al., 2017; Korolev et al., 2003) with downstream impacts on cloud evolution and lifecycle and present challenges for parameterization in models (Tan et al., 2016). Clearly, it is critical to provide improved observations of the vertical and horizontal structure across the diverse
range of mixed phase environments, to improve our understanding of variability and distribution of cloud phase at the sub-cloud-system scale.

Ground-based, airborne, and spaceborne active remote sensing techniques offer a valuable capability for understanding mixed phase cloud structure through vertically resolved cloud profiles. Combined radar-lidar methods have been used to leverage the typical size difference between supercooled water droplets (enhanced lidar backscatter) and ice particles (enhanced radar
reflectivity), often with thresholds used to mask different regions (e.g., Shupe, 2007) or other clustering methods (e.g., Romatschke and Vivekanandan, 2022). Lidar has been used extensively for cloud detection, with polarization lidar specifically used to identify ice clouds using the distinct depolarization signature (Sassen, 1991). Airborne lidar offers a unique vantage point from which to observe the three-dimensional arrangement of liquid- and ice-containing layers and regions within complex cloud systems. However, in many scenarios relevant to airborne and spaceborne remote sensing, dense water clouds
also generate depolarization through photon multiple scattering (Platt et al., 1999), which complicates a simple independent attribution of depolarization to ice. In addition, the diversity of ice particle types also results in a range of ice-only linear depolarization ratios (Okamoto et al., 2019, Noel et al., 2004).

In a recent analysis of Southern Ocean clouds, Mace et al. (2021) suggested that satellite lidar-based estimates of mixed phase clouds may be underestimated outside of convective regions where, they claim, ice multiplication processes and lofting to
cloud tops makes the ice more visible. One challenge with downward looking lidar systems in these environments is the inability to probe deep into the cloud layer; compounding this is the fact, that in many scenarios, ice is optically insignificant within the mixture. It does, however, motivate the need to better connect cloud phase classification algorithms and thresholds with physical quantities that are relevant to microphysical processes, budgets, and climate impacts, because, at some limit, alleged mixed phase clouds are essentially supercooled water layers with negligible ice contents.

Here we develop a method to utilize advanced high-electrical-bandwidth airborne lidar systems to observe fine-scale variability in the distribution of cloud phase using polarization. We seek to understand the capability of airborne lidar to discriminate ice and water in dense clouds and use collocated in situ measurements to quantify the microphysical definition of our phase categorization. The paper is organized as follows: a new method for separating ice depolarization enhancements from signatures generated from multiple scattering is developed in Section 2. In Section 3, case studies are described that involve a
second aircraft sampling marine clouds using in situ probes and synchronized to the lidar observations. Section 4 provides




quantitative evaluation of the retrieved phase vertical distributions, aided by the in situ measurements. A summary is provided in Section 5.

## 2 Methods

### 2.1 Preliminaries

#### 2.1.1 NASA Langley Airborne Lidar Systems

The cloud phase retrieval was developed for two NASA Langley Research Center airborne high spectral resolution lidar (HSRL) systems: the second generation airborne High Spectral Resolution Lidar (HSRL-2) and the High-Altitude Lidar Observatory (HALO). Both systems include HSRL capability at 532 nm and elastic backscatter at 1064 nm with polarization detection (Hair et al., 2008). HSRL-2 also includes 355 nm HSRL measurements with polarization detection (Burton et al.,
2018) and HALO includes a differential absorption lidar that can be configured for retrieval of methane (Barton-Grimley et al., 2022) or water vapor (Carroll et al., 2022). The current method utilizes the 532 nm channels common to both instruments, which have been optimized for dense cloud sampling.

#### 2.1.2 Airborne Field Campaigns

Airborne field campaign datasets include the Aerosol Cloud meTeorology Interactions oVer the western ATlantic Experiment
(ACTIVATE; Sorooshian et al., 2023) and the Convective Processes Experiment – Cabo Verde (CPEX-CV; Nowottnick et al., 2024). ACTIVATE was conducted during 2020-2022 based at NASA Langley Research Center and Bermuda and included HSRL-2, while CPEX was based out of Sal, Cabo Verde during summer 2022 and included HALO.

#### 2.1.3 High Spectral Resolution Lidar Technique

Full details of the NASA Langley HSRL technique and calibration are provided elsewhere (Hair et al., 2008). Briefly, the
HSRL isolates the spectrally broadened molecular backscatter from the total backscatter, which also includes aerosol and cloud particles. This is achieved by operating two detector channels on the co-polarized backscattered light that discriminate the signal using an iodine filter (or a Michelson interferometer at 355 nm). In doing so, (i) the lidar calibration simplifies to the determination of a channel gain ratio, because retrievals only rely on signal ratios, (ii) the particle backscatter coefficient can be determined with few additional assumptions because attenuation affects both channels equally, and (iii) the molecular
channel can be used to retrieve particle extinction independently from backscatter.

Outside the overlap region, expressions for the lidar signals corrected for range (r), channel gains ($G_m$, $G_{dep}$) and the atmospheric-state-dependent filter transmission (F) are shown in Equations 1-3, corresponding to the co-polarized total and molecular channels ($P^{\parallel}$, $P^{\parallel}_m$) and cross-polarized channel ($P^{\perp}$), respectively. The scattering ratio (SR), defined as the ratio of particle backscatter coefficient ($\beta_p = \beta_p^{\parallel} + \beta_p^{\perp}$) to molecular backscatter coefficient ($\beta_m = \beta_m^{\parallel} + \beta_m^{\perp}$), is calculated using
Equation 4 and the volume depolarization ratio ($\delta_v$) is obtained from Equation 5.

$$X^{\parallel} \equiv P^{\parallel} r^2 = c\left(\beta_m^{\parallel} + \beta_p^{\parallel}\right) T^2 \qquad (1)$$

$$X_m^{\parallel} \equiv \frac{P_m^{\parallel} r^2}{F} G_m = c\beta_m^{\parallel} T^2 \qquad (2)$$

$$X^{\perp} \equiv \frac{P^{\perp} r^2}{G_{dep}} = c\left(\beta_m^{\perp} + \beta_p^{\perp}\right) T^2 \qquad (3)$$




$$SR = \frac{X^{\parallel}+X^{\perp}}{(1+\delta_m)X_m^{\parallel}} - 1 \quad (4)$$

$$\delta_v = \frac{X^{\perp}}{X^{\parallel}} \quad (5)$$

In Equations 1-3, the raw lidar signals (P) and corrected signals (X) depend on a lidar system calibration (c) and the atmospheric transmission (T), but these terms cancel in Equation 4 and 5. The molecular depolarization ratio ($\delta_m$) is set at $\delta_m$ =0.0035 and assumed constant.

### 2.1.4 Cloud Top Identification

While there is no universal optical definition for clouds, here tops were identified as the last upward crossing of SR=10 prior to the first upward crossing of SR=50, examined along a nadir profile. The SR>50 requirement ensures that the profile reaches and exceeds a cloud extinction of the order 1 km$^{-1}$ and by retreating upwards from this threshold to SR=10, the definition of the cloud incorporates filaments near the edge, while remaining elevated from the surrounding aerosol background. Although suitable for the datasets investigated here, these thresholds may require adjustment for clouds embedded within more enhanced aerosol layers.

### 2.1.5 Lidar Signals in Clouds

The backscatter coefficient in dense clouds can exceed the backscatter associated with typical aerosol layers by three to four orders of magnitude. Under these conditions, a sufficiently high contrast may not be achieved to ensure the molecular channel remains completely free from particle contributions. Moreover, the molecular signal quickly decays causing the SR to become increasingly susceptible to noise (Equation 4).

Instead, we take an alternative approach of utilizing the HSRL signal ($X^{\parallel}_m$) only above the cloud. The atmospheric transmission at a given cloud depth, z, comprises the component between the lidar and the cloud ($T_{norm}$), the transmission in the cloud due to molecules ($T_m$; assumed known), and the transmission due to cloud particles, $T_{p,mult}$ (Equation 6). $T_{p,mult}$ incorporates the apparent reduction in attenuation as a result of multiple scattering. Rearranging Equation 2 and averaging over a normalization window (assigned to be 100 m) just above the cloud top results in Equation 7, which can then be used to generate expressions for the co- and cross-polarized cloud-attenuated backscatter coefficients (Equations 8 and 9)

$$T = T_{norm} \times T_m \times T_{p,mult} \quad (6)$$

$$cT_{norm}^2 = \langle \frac{X_m^{\parallel}}{\beta_m^{\parallel}} \rangle_{norm} \quad (7)$$

$$\beta_{atten}^{\parallel} = (\beta_m^{\parallel} + \beta_p^{\parallel})T_{p,mult}^2 = \frac{X^{\parallel}}{T_m^2 \langle \frac{X_m^{\parallel}}{\beta_m^{\parallel}} \rangle_{norm}} \quad (8)$$

$$\beta_{atten}^{\perp} = (\beta_m^{\perp} + \beta_p^{\perp})T_{p,mult}^2 = \frac{X^{\perp}}{T_m^2 \langle \frac{X_m^{\parallel}}{\beta_m^{\parallel}} \rangle_{norm}} \quad (9)$$





Analogous to the standard HSRL backscatter retrieval (Equation 4), this method incorporates a lidar calibration (by cancelling c) and employs a known state-dependent profile for $\beta_m$. The co- and cross-polarized integrated attenuated backscatter components can be written:

$$\gamma^{\parallel}(z) = \int_0^z \beta_{atten}^{\parallel} \, dz \qquad (10)$$

$$\gamma^{\perp}(z) = \int_0^z \beta_{atten}^{\perp} \, dz \qquad (11)$$

such that $\gamma = \gamma^{\parallel} + \gamma^{\perp}$. The volume depolarization ratio remains unaffected by the normalization (Equation 12), and the layer integrated depolarization ratio (D) is weighted by the backscatter and can be viewed as a mean depolarization ratio, if $\gamma^{\parallel}$ is used as a depth coordinate (Equation 13).

$$\delta_v(z) = \frac{X^{\perp}}{X^{\parallel}} = \frac{\beta_{atten}^{\perp}}{\beta_{atten}^{\parallel}} \quad (12)$$

$$D(z) = \frac{\gamma^{\perp}(z)}{\gamma^{\parallel}(z)} = \frac{\int_0^{\gamma^{\parallel}(z)} \delta_v \, d\gamma^{\parallel}}{\int_0^{\gamma^{\parallel}(z)} d\gamma^{\parallel}} \quad (13)$$

### 2.2 Multiple Scattering Depolarization (MSD) Retrieval

Figure 1a shows a generic forward model framework (black arrows) for lidar observations of dense water clouds where the inputs are the extinction profile, the droplet size distribution (microphysics), and details of the lidar system including the transmitter and receiver optics. Inversions are challenged by a combination of suitably modelling the multiple scattering (e.g., by using Monte Carlo simulations; Hu et al., 2001) without prohibitive computational cost and the fact that, in many cases, the microphysics and structure of the cloud are under constrained (Sassen and Zhao, 1995). Reduced order models and lookup tables for multiple scattering (Malinka and Zege, 2007; Donovan et al., 2015) and more information gained from multiple field-of-view systems (e.g., Schmidt et al., 2013, Pounder et al., 2012; Jimenez et al., 2020) may alleviate the problem for inversions using optimal estimation (Wang et al., 2022; Donovan et al., 2015). Hu et al. (2006) found that for clouds containing spherical water droplets, the relationship between depolarization and the integrated attenuated backscatter enhancement was essentially independent of the cloud properties and the lidar geometry (red arrow), so that measured accumulated depolarization through a cloud layer could be used to correct for multiple scattering (Roy and Cao, 2010) and this inversion method (red dashed arrow) could be independent of the lidar system.

The use of the convenient Hu relationship is no longer appropriate if there are other mechanisms causing depolarization, as is assumed to be the case with ice-containing clouds. However, by making a rudimentary, system dependent estimate of the enhancement in attenuated backscatter attributed to multiple scattering then a simplified hypothetical liquid water cloud extinction profile can then be used to model the system dependent multiple scattering depolarization (MSD) profile (Figure 1b). The measured depolarization profile can then be compared to the modelled MSD to predict the vertical distribution of cloud phase, forming the cornerstone of the method. Worth noting is that, in contrast to previous approaches (Hu et al., 2006; Hu, 2007; Roy and Cao, 2010), these depolarization profiles are local rather than accumulated from cloud edge, which is deemed more suitable for identifying gradients in cloud phase.

For emphasis, the conceptual difference between the two approaches shown in Figure 1 is that in (a), by assuming or using prior knowledge that the cloud contains exclusively liquid droplets, the measured accumulated depolarization is leveraged as a universal predictor of multiple scattering attenuated backscatter enhancement, while in (b) system-dependent knowledge is





used to estimate the depolarization generated by a hypothetical water cloud that produced the observed attenuated backscatter profile.

### 2.2.1 Examination of Water-Only Clouds

Clouds that were sufficiently warm to guarantee a liquid phase were used to evaluate the lidar observations across a range of operating conditions. Four water cloud control cases were investigated spanning the range of viewing geometry expected from most airborne operations (Table 1) and included cases from both HALO and HSRL-2. The viewing geometry is predominantly affected by the range-to-cloud (RTC) because both systems have the same HSRL transmitter/receiver architecture (e.g., 1 mrad field of view with a transmit/receive geometry).

Control case CPEX comprised a low stratocumulus cloud located over the eastern tropical Atlantic near the coast of Africa, control case ACTLOW was a widespread stratus deck situated over the cool coastal waters between the Gulf Steam and the Atlantic seaboard of the United States, and control case ACTHIGH was an altocumulus cloud associated with a weak frontal boundary over the western North Atlantic. ACTLOW1 and ACTLOW2 were two crossings of the same cloud region separated by about 4 hours. The GRD control case occurred during ground calibration and testing in the laboratory and captured a region of stratus ahead of a warm front (i.e., with the airborne lidar facing zenith). These warm cloud control cases were selected
because the clouds were mostly opaque, assumed to be well mixed, and with relatively uniform tops.

Except for GRD, control cases were further screened to isolate individual profiles that are sharp edged with high extinction, near-adiabatic, opaque signatures that are collectively determined by (i) complete attenuation of the lidar, (ii) above-median peak $\beta^{\parallel}_{atten}$, and (iii) below-median peak $\beta^{\parallel}_{atten}$ rise depth. Across all control cases, screened statistics of the integrated quantities (Figure 2a) show that an increase in integrated backscatter is associated with an increase in integrated depolarization ratio. The
control cases closely conform to the expected relationship between the enhancement of integrated attenuated backscatter and depolarization (Hu et al., 2006) with a cloud lidar ratio ($S_c$) set to a suitable value for non-precipitating water clouds ($S_{c,ref}$ = 19 sr). As the RTC increases, the integrated attenuated backscatter increases (Figure 2b) compared to an opaque single scattering reference value ($\gamma^{\parallel}_{ref} = 1/2 S_{c,ref}$) after Platt (1973).

### 2.2.2 Extinction Inversion

An estimate of the extinction is a necessary intermediate step for the MSD retrieval and Equations 14-16 show the steps taken to calculate an estimated extinction profile ($\alpha^*$). The result of calculating $\alpha^*$ should be viewed as the extinction profile of a hypothetical water cloud with $S_c = S_{c,ref}$ that produces the measured $\gamma^{\parallel}(z)$ and not necessarily an estimate of the true extinction. As an example, $\alpha^*$ would underestimate the true extinction profile for conditions where $S_c > S_{c,ref}$ that may occur with sufficient ice content; however, as will become more apparent, the underestimate usually benefits rather than hinders ice
discrimination.

The value of $\gamma^*$ is set to the maximum value of the $\gamma^{\parallel}(z)$ profile and serves as a normalization coefficient for the subsequent extinction calculation. In addition, a lower bound threshold on $\gamma^*$ was prescribed using a predetermined, RTC dependent, opaque cloud value ($\gamma^{\parallel}_{rtc}$) that was generated from a linear fit through the minima (10th percentile) of the nadir water cloud control case data (Figure 2b). It is important that $\gamma^*$ be at or above the maximum of $\gamma^{\parallel}(z)$ to ensure that Equation 16 provides
physical solutions. When the profile maximum is above $\gamma^{\parallel}_{rtc}$, it typically implies an opaque profile, and the surplus is caused by the combination of a lower $S_c$ and/or more multiple scattering than the control case. Capping the lower limit at $\gamma^{\parallel}_{rtc}$ partially circumvents problems that arise with translucent clouds, where further constraints on the lower boundary condition for the cloud are needed (e.g., Young, 1995). Opaque clouds do not need limits placed on $\gamma^*$, so it is justifiable to establish $\gamma^{\parallel}_{rtc}$ using the minimum levels of multiple scattering enhancement from the control case data.





S* incorporates the multiple scattering effects into a column effective lidar ratio, but in cases with ice we lack knowledge of how the multiple scattering builds through the profile to employ the method of Roy and Cao (2010). However, we can leverage the boundary condition at the top of the cloud where single scattering prevails, $\gamma^{\parallel} \to 0$ and $T^2_{p,mult} \to 1$, which results in the scaling factor, $S_{c,ref}/S^*$, applied to Equation 16. A novel aspect of this approach is that the co-polarized signal was used rather than a total integrated backscatter because the cross-polarized backscatter from spherical droplets only arises from multiple

scattering, while the co-polarized component results from both single and multiple scattering. Since the single scattering properties are needed to estimate the extinction, retaining the cross-polarized component provides no benefit and requires a larger correction for multiple scattering that may result in a greater uncertainty. For implementation of the method, $\Delta z$ is allowed to vary to keep increments in $\gamma^{\parallel}$ constant (up to the native acquisition of the measurement $\Delta z$ = 1.25 m) and this has the advantage of mitigating the decrease in signal-to-noise ratio with depth in the cloud that affects both $\alpha^*$ and $\delta$.


$$\gamma^* = \max\left[\gamma^{\parallel}{}_{max}, \gamma^{\parallel}{}_{rtc}\right] \quad (14)$$

$$S^* = \frac{1}{2\gamma^*} \quad (15)$$

$$\alpha^*(z) = \frac{-\Delta \log\left(1 - 2S^* \gamma^{\parallel}(z)\right)}{2\Delta z} \frac{S_{c,ref}}{S^*} \quad (16)$$

### 225  2.2.3 MSD Empirical Model

Primed with an estimate of the extinction profile, we can now approach the task of predicting the MSD profile produced by a hypothetical water cloud. To do this we will utilize the nadir-facing warm cloud control cases, but first we need to establish a framework to empirically model the MSD. In anticipation that the MSD profile responds to the accumulated cloud properties along the path, we choose to base the model framework on a first order ordinary differential equation (ODE) of the form:

$$\delta'(z) = f\big(\delta(z), \alpha(z), \alpha'(z)\big) \quad (17)$$

One potential organizing framework is to express the gradient in depolarization as summations of a relaxation term ($\delta' \sim -\delta$), an accumulation term ($\delta' \sim \alpha^b$), and a proportional term ($\frac{\delta'}{\delta} \sim \frac{\alpha'}{\alpha}$). The relaxation term provides a way to capture both the physical upper limit of the depolarization ratio and the progressive loss of scattered photons from the field of view. The accumulation term captures the growth of depolarization with optical depth and recognizes the fact that more extinction creates

a more rapid increase (b>0). The proportional term aims to capture the fact that rapid fractional changes in the extinction may physically create depolarization gradients that are otherwise not mathematically captured by the other terms. This was deemed important to include in the model based on the observed characteristics of multilayered clouds and where a marked step in the extinction was embedded within a single cloud layer. Combining these terms results in the following expression ($\alpha = \alpha(z)$ and $\delta = \delta(z)$ are implicit for brevity):

$$\delta' = -r_1 \delta + r_2 \alpha^b + k\frac{\alpha'}{\alpha}\delta \quad (18)$$

where $k = k_+$ for $\alpha' > 0$ and $k = k_-$ for $\alpha' < 0$ to allow different physical processes associated with increasing and decreasing extinction gradients to be captured. The five free parameters ($r_1$, $r_2$, b, $k_+$, $k_-$) remain to be optimized but are assumed to be model constants, the extinction profile is the input, and the output is the MSD profile. The model was solved numerically





along the beam path using a simple implicit Euler method using increments $\Delta z$, initial condition $\delta_{i=0} = 0$, and the estimated
extinction profile, $\alpha^*(z)$:

$$\delta_{i+1} = \frac{\delta_i + \Delta z r_2 \alpha^*_{i+1}{}^b}{1 + \Delta z r_1 - k(\alpha^*_{i+1} - \alpha^*_i)/\alpha^*_{i+1}} \quad (19)$$

with $k = k_+$ for $\alpha^*_{i+1} > \alpha^*_i$ and $k = k_-$ for $\alpha^*_{i+1} < \alpha^*_i$

### 2.2.4 Determining Optimal Model Parameters

Optimal model parameters ($r_1$, $r_2$, $b$, $k_+$, $k_-$) were those that minimize the root-mean-squared error (RMSE) between the
measured depolarization and modelled MSD profiles for the nadir-facing control case datasets (Table 1) and are shown in Table 2. Although the training sets were independent, there was consistency in the relationships amongst the parameters. Of note was the consistency in $k<0$ capturing a decrease in depolarization at an upward gradient in extinction, perhaps caused by a temporary increase in the relative importance of single scattering. The opposite effect (i.e., an increase in depolarization when extinction decreases) was a feature of multiple scattering simulations of multi-layered clouds (Roy and Tremblay, 2022).

An increase in the $r_2$ accumulation rate term with RTC was expected based on the behaviour of integrated values (Figure 2a) and a suitable set of final values could be achieved by holding the other parameters fixed within a region of overlap (i.e., relaxing the requirement to be held at the depolarization RMSE minimum). The final parameters (Table 2) trade a small penalty in RMSE for model simplicity and are found satisfactory for the current usage, acknowledging that improvements may be possible with additional training sets.

### 2.2.5 Modelled and Measured MSD

Figure 3 shows example comparisons between the measured depolarization profile and the predicted MSD, following the above steps. The $\beta^{\parallel}_{atten}$ profiles (Figure 3a,d,g) are used to generate $\alpha^*$ profiles (Figure 3b,e,h) and then the MSD model produces predicted $\delta$ profiles that are compared to the measured $\delta$ profile (Figure 3c,f,i). The first profile (Figure 3a-c), chosen from the CPEX control case, shows a relatively constant extinction (after the first 15 m) and a depolarization profile that grows
monotonically. The second profile (Figure 3d-f) is taken from the ACTLOW1 control case and has a marked step up in the extinction profile at a depth of approximately 40 m with an associated temporary decrease in depolarization. The third profile (Figure 3g-i) shows a translucent cloud record taken from ACTHIGH and in this case the MSD model overestimates the $\delta$ and it may indicate that translucent clouds and/or lower RTC conditions have been penalized by the compromise in the choice of final parameters in Table 2. However, the MSD is generally lower for lower RTC making ice discrimination under these types
of conditions less ambiguous.

In each example, the sensitivity to a ±5% perturbation in the value of S* is evaluated with respect to its impact on $\alpha^*$ and predicted $\delta$. The $\alpha^*$ sensitivity is negligible near cloud top but grows very rapidly with depth in the first two cases, ultimately resulting in the retrieval diverging (and becoming undefined) because of negative transmission in the case of positive S* anomalies. However, until the point where the profile becomes undefined, the MSD model prediction is comparatively less
sensitive. This quality motivates the approach of calculating the extinction as an intermediate step, despite the simplistic multiple scattering assumptions.

In contrast, and as an introduction to the next section, Figure 4 shows three example profiles taken from a supercooled cloud cluster (< 25 km separation) sampled as part of ACTIVATE during a cold air outbreak event on March 29, 2022. The format of Figure 4 is the same as Figure 3 but here the measured $\delta$ profile (Figure 4c,f,i) diverges from the MSD model, clearly
highlighting the regions of the profile where ice can be identified. In the first example (Figure 4a-c), there is a consistent enhancement in the measured $\delta$ suggesting a mixed phase profile, while in the second example (Figure 4d-f) the profile is initially similar to the translucent water cloud (Figure 3g-i) but after approximately 100 m, there is a marked increase in $\delta$





indicative of a transition to ice-dominant conditions. The third example (Figure 4g-i) shows the opposite configuration with an ice-dominant signature occurring atop a liquid-dominant layer.

## 2.3 Identification of Ice-Containing Layers

### 2.3.1 Phase Mask

If a region within the cloud has a depolarization ratio that significantly exceeds the MSD profile, then it is assumed to contain randomly oriented, irregular ice (Yoshida et al., 2010; Hu et al., 2007; Hu et al., 2009; Mace et al., 2020). Here, the measure of significance is a buffer region around the MSD profile accounting for measurement error and uncertainty in the empirical model, primarily to minimize false positives in warm clouds (Figure 5). The buffer region was set to exceed the MSD by 10% plus an additional 0.06, which kept the threshold above the 95[th] percentile across the range of MSD and also maintained an overall false positive frequency between 0.48% and 2.2% depending on the control case.

Measurements that fall under this threshold are essentially indistinguishable from water-only clouds (WATER category), while those lying above are attributed to either mixed phase (MIX category) or conditions dominated by randomly oriented, irregular ice (ICE category). The distinction between MIX and ICE is set at a depolarization ratio of 0.35, based on airborne HSRL measurements of ice particle depolarization at low SR (Burton et al., 2012). When the MSD>0.35, no MIX category exists and MSD defines the boundary between WATER and ICE, recognizing that identifying ice in these scenarios is likely to be highly ambiguous (Yoshida et al., 2010).

Regions containing pristine oriented ice is another potential scenario, where crystal facets produce non-depolarizing, strong specular reflections (Platt 1978). Hu et al. (2007) explored quantifying varying contributions of oriented ice using an end member with very low lidar ratio and depolarization. Here, if the signature of oriented ice is strong enough, the extinction, estimated using water cloud assumptions, will be overpredicted artificially boosting the MSD, while the observed depolarization ratio will tend to decrease. Therefore, if the depolarization ratio drops significantly below the MSD profile, then this region is assumed to contain oriented ice (HOI category). Additionally, if the integrated backscatter exceeded the maximum water cloud backscatter enhancement expected for that RTC (Figure 2b) then regions within the profile defined as WATER were replaced with HOI.

The mixed phase cloud environments studied here were not expected to be dominated by pristine ice because of the expectation for rime (Chellapan et al., 2024), which may enhance ice depolarization (Sassen, 1991), but a method to flag any oriented ice was necessary to avoid misclassification in the WATER category. With this method, distinct subregions and layers categorized as ICE or HOI are usually obtainable; however true mixtures containing oriented ice in varying proportion may result in some level of misclassification as WATER or MIX.

### 2.3.2 Auxiliary Categories

Regions below cloud top that are otherwise classified as WATER but have low extinction are recategorized as non-cloudy mainly to aid in situ validation. If these regions were retained, it would likely bias any statistical comparison because they no longer meet the definition of a water cloud. These regions are categorized as DIM to highlight that the low backscatter could result from a cloud free zone or potentially an underprediction of S*.

The threshold defining cloud top may be too high to include weak ice signatures that are otherwise useful for providing additional context. If the region above the cloud has a volume depolarization above 0.2, these regions are categorized as δABOVE and the use of volume depolarization means that low SR regions are automatically filtered out. These auxiliary categories provide context but are not used in the quantitative validation.



## 3 Results and Testing

### 3.1 ACTIVATE Flight Strategy

During ACTIVATE, a unique sampling strategy was employed where a remote sensing turboprop aircraft (King Air B200 or UC12, henceforth King Air) was flown at 8-9 km in coordination with a low-flying in situ platform (Dassault HU-25 Falcon,
henceforth Falcon) that operated at multiple altitudes to sample aerosols and clouds. The use of these specific aircraft and operating altitude ranges allowed both to remain approximately matched in flight speed and offer an unparalleled dataset for remote sensing validation.

### 3.2 Aircraft Coincidence

Even though the two aircraft were well suited to remain speed-matched over long transects, in practice there was often some
deviation from perfect coincidence and usually this related to a small difference in the time at which each aircraft passed over a point along a common flight track. There were also some flights where relative deviations in flight track occurred, either planned or unplanned. Therefore, a coincidence algorithm was developed to allow the mapping of Falcon datasets onto the King Air, which served as the reference. For each timestamped King Air position, a centered 30-minute window (i.e., +/- 15 minutes) of Falcon data was searched for the optimal match. In this case, optimal is defined as the minimum linear distance
after implementing a linear advection approximation to account for the wind drift over the time differential. Wind profiles used for the advection are estimated using linear interpolation between dropsonde vector wind profiles. In the limiting case of perfect aircraft collocation, the wind correction is irrelevant, and the match point is synchronized in time; however, when there is a time lag, this method incorporates the lateral offset that would accumulate in the case of the Falcon experiencing crosswinds or the additional adjustment in the match point that would be needed to account for headwind or tailwind components. It was
found that, despite the tight constraints on collocation, accounting for wind was advantageous for analyzing the aircraft matchups.

### 3.3 In Situ Datasets

In situ cloud microphysical properties were evaluated using the combination of a SPEC Inc. Fast Cloud Droplet Probe (FCDP; Kirschler et al., 2023) and a SPEC Inc. 2-dimensional Stereo optical array probe (2DS; Lawson et al, 2006). The FCDP and
2DS measure particle size distributions from 3-50 µm and 28.5-1465 µm, respectively. Particle images acquired by the 2DS were used to discriminate liquid and ice particles by identifying non-spherical shapes but because the method required sufficient pixels to be occluded on the diode array, the minimum size was 90 µm for phase determination.

For this study, four microphysical classes were defined: droplets, supercooled large drops (SLD), drizzle and ice. Droplets comprised the particles measured by the FCDP and extended to 50 µm diameter, with an underlying assumption that this size
range was exclusively liquid water and spherical, irrespective of the temperature. Number concentration and extinction were calculated by integrating the particle size distribution applying suitable cross sections according to Mie theory. The SLD class comprised the particles identified as liquid by the 2DS when the ambient temperature was below 0°C, and therefore corresponded to drop diameters > 90 µm. The drizzle class was otherwise identical to SLD for non-supercooled conditions (> 0°C), therefore these classes never occurred simultaneously. Number concentration and extinction for these classes were
calculated with the same assumptions as for droplets. Ice was limited to particles sizes >90 µm and the ice number concentration was calculated by integrating the particle size distribution of ice classified particles, while the extinction was determined using the empirical relationships provided in Platt (1997). Based on the thresholds set for the image processing, it is assumed that ice false positives were negligible (estimated at < 1%) while ice false negatives may comprise 10-15% of the data otherwise attributed to SLD and reflects an uncertainty in the data classification.

The number concentration and fractional extinction of each class were determined at 1 Hz and to summarize, the following caveats should be highlighted: (i) very small ice <50 µm, if present, would be misclassified as cloud droplets, (ii) the region



of the particle size distribution 50-90 µm is completely ignored because phase discrimination is ambiguous with the available instruments, (iii) estimates of ice particle extinction are expected to carry more uncertainty.

### 3.4 Cloud Scene Identification

Only a small fraction of the total ACTIVATE data sampling involved the Falcon flying near cloud top, even though a large fraction of flights took place with cloudy conditions. Furthermore, many of the clouds were not supercooled or did not contain SLD or ice and therefore were not useful cases to evaluate cloud phase distributions. Each of the 162 joint flights were reviewed to evaluate flight segments where the matched Falcon data involved sampling of cloudy regions containing varying distributions of droplets, SLD and ice that were also observable by HSRL and Table 3 shows the details of the segments that
were reserved for further analysis. Segments were 5-24 minutes in duration, Falcon average cloud temperatures varied from -13˚C to -3˚C and the largest time offset was approximately 5 minutes.

### 3.5 Example 1: February 28, 2020

Example 1 comprised a transect across a region of aggregated shallow convection associated with a deep unstable marine boundary layer (Figure 6). In the center of the cloud scene, active convective cells resulted in locally higher cloud tops, while
the surrounding regions contained surface-decoupled stratiform layers that also extended beyond the boundary of the segment. The Falcon transect involved ascending and descending altitude ramps bracketing a constant altitude leg crossing the deepest region of the cloud. The initial ascent through the cloud (at 10-12 km along track, Figure 6b,c) exclusively involved small liquid droplets corroborating the HSRL retrieval in this region. After a brief period above, the Falcon re-entered the rising cloud tops and encountered mixed phase conditions with varying influence from ice and supercooled drops (18-48 km along
track) before reverting to liquid only conditions after 52 km. While the track of the Falcon for these regions was often deeper in the cloud than the extinction limit of the lidar, the horizontal representation of the ice containing regions was certainly captured and in agreement.

### 3.6 Example 2: April 2, 2021

Figure 7 shows a multi-layered cloud structure comprising a broken stratiform upper layer with tops near the inversion base at
3 km and a second layer with cumulus cloud tops mostly around 2.3 km. In two regions (verified using onboard camera imagery) the cumulus clouds were more vertically developed and coupled the otherwise distinct layers at 50-60 km and 70-90 km along track, respectively. The Falcon transect involved an ascent profile up to an above cloud top level leg followed by an in cloud level leg nominally just below the tops. Similar to Example 1, the initial climb involved a penetration through liquid dominant cloud (10-14 km along track) and verified both the aircraft matchup and HSRL phase identification in this
region. Descending into cloud, the Falcon encountered a mixed phase environment with subregions of ice with some SLD punctuated by liquid-dominant cores. Although retrievals were not always available at the Falcon altitude, the location of the liquid-dominant cores can be traced to regions of dense water cloud above, identified by HSRL. This is particularly evident with the cores observed at 85 km and 89 km along track where the agreement in the spatial position of the phase discrimination is captured with excellent fidelity.

### 3.7 Example 3: January 27, 2022

In this example, cloud tops were quite uniform and mostly constrained to the 1.5-1.7 km range for the first 40 km, where the cloud coordinated sampling took place (Figure 8). The dropsonde release used to constrain the temperature profile (Figure 8a) occurred at the end of the segment, located over the Gulf Stream, and may explain the increase in cloud tops and the increase in the inversion base (2.3 km, not shown). Another dropsonde was released approximately 250 km prior to the start of the
segment over cooler waters with an inversion base marking the boundary layer top at 1.5 km and -8.9˚C and may be a closer representation of the profile near start of the segment despite the greater distance.





This example represents a significant challenge for the lidar because the cloud top region is assumed to be dominated by water extinction making any ice difficult to identify. The Falcon sampled 200-300 m below the cloud tops (first 40 km along track) and observed a slowly varying background of ice and SLD with more rapid variability in the droplet number. This pattern is
interpreted as the aircraft transecting numerous embedded liquid-dominant mixed phase cells, causing the cloud phase fraction to be mostly driven by the liquid variability. From the observed attenuation of the lidar, it can be inferred that these liquid-dominant cells spread laterally near the inversion, obscuring the view of most of the structural variability observed by the Falcon. The exception was one region (20-25 km along track) where the liquid-dominant cloud tops became more tenuous allowing the lidar to penetrate deeper and observe the presence of ice at the Falcon altitude. This structure was also captured
in the Falcon data with the more extensive high ice extinction fraction measured in that region. A further notable feature of Example 3 is the ubiquity of more pristine column ice (Figure 8d) compared to the rimed and aggregated particles observed in the previous two examples. Specular reflections from ice facets may tend to produce lower depolarization ratios for these ice crystals compared to the other cases, potentially explaining the predominance of the MIX category instead of ICE in high ice extinction fraction regions. Therefore, the natural variability in the depolarization caused by different crystal habits and particle
growth mechanisms (e.g., riming) highlights some of the limitations with a depolarization threshold for discriminating MIX and ICE categories.

## 4 Validation

At each collocated cloud scene, the HSRL vertically resolved cloud phase was compared to the cloud microphysics measured by the Falcon. While the qualitative interpretation discussed for the three examples demonstrates the utility of the cloud phase
retrieval, there are many possible methods to assess the skill of the retrieval within the constraints of the sampling strategy. At the core of the retrieval validation is the fact that aircraft collocation errors remain comingled and neither HSRL nor Falcon measurements provide an unbiased statistical representation of the cloud scene's vertical cross section. The Falcon was limited to its flight altitude, which varied relative to the local cloud topography even during level legs. The HSRL penetration depth varies inversely with the extinction, such that dense layers at cloud top, that are usually liquid dominant, obscure lower levels
where the cloud phase distribution may be different. Furthermore, the HSRL phase classification does not directly translate to the Falcon's microphysical measurements and therefore it is sensitive to choices for thresholds on both the definition of cloud and the amount of ice required to be classified as mixed phase.

To minimize bias, each scene was screened to reject records, for both platforms, where there was no overlap with HSRL phase data vertically within a 300 m zone above and below the Falcon. This vertical scale was selected as a compromise between
minimizing potential collocation errors and filtering out too much data. The ice thresholds bounding the mixed phase classification for the in situ data were left as adjustable parameters, with the logic being that they could be adjusted to better define the bounds of the HSRL MIX category. An ice mixing fraction was defined:

$$\mu_{ice} = \frac{\alpha_{ice}}{\alpha_{ice} + \alpha_{SLD} + \alpha_D} \quad (20)$$

where subscripts ice, SLD and D (droplets) refer to the in situ extinction fractions defined earlier. The in situ definition of
cloud was set to 0.05 km$^{-1}$ and an in situ record was counted as containing ice if:

$$\mu_{ice} > \mu_{thresh} \quad (21)$$

Each cloud scene was divided into subsegments with length Δx (a tuneable parameter) with overlap such that sequential subsegments were oversampled with offset Δx/4. The in situ frequency of ice-containing cloud was calculated as the ratio of data points that satisfy Equation 22 to the total number of cloudy points within the subsegment, subject to the vertical
collocation screening. The frequency of HSRL ice-containing cloud phase retrievals was similarly calculated by counting range bins classified as MIX, ICE and HOI across all qualifying records within the segment divided by the total cloud classified bins. Together these frequencies indicate the statistical probability of the respective cloud phase classification of the two



aircraft matchup within the qualifying sub-volume. A 20% counting threshold was set for both the fraction of in situ cloud-containing points and the fraction of qualified HSRL records to reject comparison for sparsely populated subsegments. For a given Δx, each cloud scene contained a different number of suitable subsegments because of the variable length of the scene, the degree of aircraft collocation, and the specifics of the cloud morphology. If the subsegment window exceeded the scene length it was truncated and no further subsegments were examined, hence, by increasing Δx, the number of subsegments per scene reduced until all scenes contained one subsegment. A cartoon illustration of this quantitative matchup method is shown in Figure 9.

Figure 10a shows the comparison between the HSRL and in situ subsegment probabilities, with $\mu_{thresh} = 0.14$ and $\Delta x = 40$ km, and serves as a measure for validating the skill of the HSRL phase retrieval of ice-containing cloud for these scenes and aircraft collocation. The mean absolute error (MAE) was 0.12 and incorporates both retrieval uncertainty and any remaining sampling collocation errors, despite efforts to minimize them. The procedure was repeated (Figure 10b), adjusting the HSRL classification to only include ICE and HOI (i.e., disallowing the MIX category) and altering the comparison to $\mu_{thresh} = 0.76$, which examines the skill of the HSRL phase retrieval for identifying ice-dominant cloud. Here the slightly lower MAE=0.09 is attributed to the reduced dynamic range offered by these cases, since few scenes contained significant ice-dominant segments. The few segments with any influence from HOI classifications are highlighted by the magenta border in Figure 10a,b (HOI > 2%). As previously mentioned, oriented ice creates an additional complication for this method, which is primarily designed to assess binary mixtures of water droplets and depolarizing irregular ice. If the HOI category was not differentiated from WATER, particularly for Scene 6, the retrieval skill would have been unjustifiably penalized.

The sensitivity to the parametric choice of $\mu_{thresh}$ was examined by evaluating the MAE for ice-containing and ice-dominant comparisons across the full range (Figure 10c) with fixed $\Delta x = 40$ km. A higher density of computations was conducted in the regions near the optimal thresholds, which correspond to the aforementioned selections for Figure 10a,b. Also shown for reference are the values of the correlation coefficient (R), which was considered as a secondary metric for performance in addition to the MAE. The consequence of the sensitivity analysis is that the optimal threshold values of 0.14 and 0.76 represent the best estimate of the lower and upper bounds of the ice extinction fraction of the HSRL MIX category. Put another way, when the ice extinction is less than 14% of the total, on average the cloudy volume would be indistinguishable from liquid only conditions, while greater than 76% ice would have sufficient depolarization to satisfy the ice-only minimum. Evaluation across a greater number of scenarios is needed to determine if these lidar thresholds are consistent and whether they suitably bracket mixed phase conditions from the perspective of microphysical processes and cloud radiative effects. Additional information content from lidar wavelength dependence, combined radar-lidar, and combined active-passive strategies may further refine phase classification and improve the detection of ice in liquid dominant conditions.

The sensitivity to the parametric choice of $\Delta x$ was also evaluated by maintaining the respective $\mu_{thresh}$ at their optimal values. At low $\Delta x$, collocation errors become increasingly dominant in response to strong variability in the cloud properties at the scale of individual cloud elements (e.g., discrete convective eddies), where it is difficult to guarantee that both aircraft encountered statistically representative transects. At intermediate $\Delta x$, the collocation errors plateau but increasing segment size diminishes the dynamic range of ice frequencies, explaining the decrease in R. The increase in MAE at the highest $\Delta x$ is less intuitive but is partially attributed to the relative weight applied to the scenes because of the number of segments. Large $\Delta x$ also tends to retain low cloud fraction and less well-matched subregions because achievement of the 20% rejection thresholds is almost guaranteed for any of the scenes, which may increase MAE. Although no such constraints were imposed, it is both interesting and encouraging that the optimal $\Delta x = 40$ km was found to be consistent between the ice-containing and ice-dominant sensitivity tests. A possible explanation is that this length scale corresponds to a dominant mesoscale mode of boundary layer cloud organization that affects these cases. The fingerprint of mesoscale organization serves to maximize the spatial variability of the cloud phase distribution, which is effectively captured when the size of validation segments is close to this length scale. It should be clarified that this interpretation is a consequence of these specific cloud environments and the $\Delta x$ parameter used to validate the retrievals and is not necessarily universal. Ultimately, mesoscale gradients serve to maximize dynamic range across a cloud scene while minimizing collocation error.



## 5 Summary

A polarization-based, vertically resolved cloud phase retrieval has been developed with specific application to the NASA Langley airborne HSRL instruments. The method seeks to separate the depolarization associated with ice particles from multiple scattering, in accordance with other polarization-based phase classification algorithms. However, the novel approach of this method is to interrogate the range-resolved depolarization profile at the smallest available scales to extract additional information about the vertical distribution pertinent to mixed-phase cloud environments that is not attainable from a layer-integrated approaches.

An empirical model to describe the MSD profile from dense water clouds was established using lidar observations of verified water-only, high extinction, non-precipitating clouds at various ranges. With airborne lidar, the range to cloudy targets is sufficiently variable to modulate the influence of multiple scattering. The present empirical model is specific to the NASA Langley airborne HSRL viewing geometry, but because the model is trained using known-to-be water clouds, the approach may be trainable on other systems. Ice-containing clouds are subsequently identified as regions of the depolarization profile
that deviate substantially from the MSD profile.

The ACTIVATE field campaign employed a unique coordinated approach where a low-flying aircraft sampled marine boundary layer clouds in situ, while a high-flying aircraft that included the HSRL-2 instrument flew the same flight line aloft. Fifteen cloud scenes were evaluated from the ACTIVATE dataset that satisfied stringent requirements for aircraft coordination, the availability of collocated measurements, supercooled cloud temperatures, and observed ice and/or supercooled large drops.

Evaluation of the frequency of ice-containing and ice-dominant conditions, which represent the boundaries of a mixed phase categorization, indicated that these thresholds were most closely associated with in situ ice extinction mixing fractions of 14% and 76%, respectively. Using these threshold mixing fractions, matchup probabilistic comparisons made on 40 km subsegments of each cloud scene exhibited mean absolute error of 0.12 and 0.09, respectively.

While ACTIVATE prioritized the coordination of the aircraft horizontally for all the survey flights (nominally < 10 minutes
flight time separation), a tighter synchrony and a cloud-top-focused strategy would be needed to further untangle collocation errors from the retrieval uncertainty. The ACTIVATE cases often comprised mixed phase environments with significant riming and aggregation of ice particles leading to high depolarization ice signatures, which was advantageous to this method. While the current algorithm has some ability to identify oriented ice regions, it is not possible to untangle ternary mixtures comprising water, irregular ice, and oriented pristine ice. Indeed, mixtures involving any oriented ice component are currently
ambiguous and require additional intensive properties (e.g., wavelength dependence) to increase the dimensionality.

Nonetheless, the degree of qualitative agreement of features observed in individual cases together with the quantitative validation demonstrate the utility of this method to capture detailed, high-resolution information about ice and water distribution in complex multiphase cloud systems.

## Data Availability

All field campaign datasets are publicly available and can be found at https://doi.org/10.5067/SUBORBITAL/ACTIVATE/DATA001 (NASA/LaRC/ASDC, 2021) and https://doi.org/10.5067/ASDC/SUBORBITAL/CPEXCV_Merge_Data_1 (NASA/LaRC/ASDC, 2023)

## Author Contribution

EC and JWH developed the retrieval method. All authors contributed to experimental data collection and data analysis. EC
led the preparation of the manuscript with contributions from all authors.



**Competing Interests**

The authors declare no competing interests.

**Acknowledgements**

We acknowledge the contributions of NASA Langley Research Services Directorate and the NASA Armstrong DC-8 for the successful execution of ACTIVATE and CPEX-CV flights. This work was supported by ACTIVATE, a NASA Earth Venture Suborbital (EVS-3) investigation funded by NASA's Earth Science Division and managed through the Earth System Science Pathfinder Program Office. CV and SK acknowledge support from the German Research Foundation.

**Financial Support**

Armin Soroooshian was supported by NASA (grant no. 80NSSC19K0442). Christiane Voigt and Simon Kirschler were funded
by DFG within projects no 510826369 (ECOCON) and no 522359172 (SPP HALO) and by the European Union's Horizon Europe and SESAR programs under grant no 101114785 (CONCERTO) and grant no 101114613 (CICONIA).

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





Table 1: Water cloud control cases used to evaluate multiple scattering.

| Control Case | View | Inst. | Campaign | Date | Time (UTC) | Platform Altitude (m) | Cloud Altitude (m) | Cloud Temperature (˚C) |
|---|---|---|---|---|---|---|---|---|
| GRD | Zenith | HALO | - | 2024-02-28 | 21:06 – 21:11 | - | 1540±5 | 14 |
| ACTLOW1 | Nadir | HSRL-2 | ACTIVATE | 2022-05-05 | 13:22 – 14:14 | 8831±20 | 600±150 | 7-14 |
| ACTLOW2 | Nadir | HSRL-2 | ACTIVATE | 2022-05-05 | 17:51 – 19:49 | 8805±29 | 500±80 | 6-13 |
| ACTHIGH | Nadir | HSRL-2 | ACTIVATE | 2022-06-18 | 13:54 – 14:02 | 8904±7 | 4580±130 | 0 |
| CPEX | Nadir | HALO | CPEX-CV | 2022-09-15 | 19:05 – 20:12 | 11996±4 | 900±45 | 23 |





Table 2: MSD model coefficients determined from residual minimization of each control case dataset. The Final coefficients were compared against all control cases.

|  | CPEX | ACTLOW1 | ACTLOW2 | ACTHIGH | Final |
|---|---|---|---|---|---|
| $r_1$ | 0.048 | 0.039 | 0.038 | 0.040 | 0.039 |
| $r_2$ | 0.110 | 0.090 | 0.086 | 0.088 | $4.094 \times 10^3 RTC + 0.06449$ |
| b | 0.55 | 0.59 | 0.57 | 0.62 | 0.608 |
| $k_+$ | -0.66 | -0.39 | -0.52 | -0.34 | -0.554 |
| $k_-$ | -0.43 | -0.46 | -0.56 | -0.14 | -0.469 |
| RMSE | 0.0220 | 0.0249 | 0.0237 | 0.0242 | 0.0248 |





Table 3: ACTIVATE Cloud Scenes used for validation.

| Scene Index | Date | Flight Leg | Time | Max. lateral offset | Lag (min/max) | HSRL cloud top (5%/median/95%) | Falcon altitude [a] | T [a] |
|---|---|---|---|---|---|---|---|---|
| | | | UTC | (km) | (s) | (km) | (km) | (°C) |
| 1 | 2020-02-28 | L2 | 20:47-20:53 | 0.2 | -220 / -86 | 2.74 / 3.12 / 3.87 | 2.90 | -8.4 |
| 2 | 2021-04-02 | L1 | 13:32-13:46 | 1.4 | -69 / -20 | 1.69 / 2.88 / 3.14 | 2.81 | -8.8 |
| 3 | | | 14:19-14:31 | 0.6 | -17 / 46 | 2.79 / 3.19 / 3.40 | 2.80 | -5.9 |
| 4 | 2021-04-02 | L2 | 18:44-19:06 | 1.6 | -264 / 263 | 2.20 / 3.43 / 3.77 | 3.32 | -11.0 |
| 5 | 2021-12-09 | L1 | 14:11-14:33 | 3.6 | 145 / 267 | 1.02 / 1.64 / 1.93 | 1.37 | -3.0 |
| 6 | 2022-01-11 | L1 | 15:22-15:31 | 1.3 | 160 / 247 | 0.85 / 2.48 / 2.66 | 2.15 | -12.9 |
| 7 | 2022-01-11 | L2 | 20:46-20:52 | 0.9 | -87 / -44 | 1.64 / 1.78 / 1.93 | 1.69 | -12.6 |
| 8 | 2022-01-15 | - | 14:51-14:58 | 0.5 | -127 / -65 | 1.72 / 1.80 / 1.89 | 1.67 | -4.9 |
| 9 | 2022-01-18 | L1 | 15:08-15:32 | 2.6 | -306 / 88 | 2.08 / 2.47 / 2.77 | 2.13 | -9.8 |
| 10 | 2022-01-19 | L1 | 15:12-15:19 | 0.7 | 104 / 166 | 2.13 / 2.18 / 2.21 | 1.94 | -3.6 |
| 11 | 2022-01-27 | L1 | 14:55-15:04 | 0.5 | 121 / 234 | 1.47 / 1.56 / 1.61 | 1.43 | -13.6 |
| 12 | 2022-01-27 | L2 | 19:46-19:57 | 0.1 | -21 / 22 | 1.48 / 1.64 / 1.72 | 1.33 | -3.6 |
| 13 | 2022-02-26 | L1 | 14:49-15:12 | 1.5 | -8 / 196 | 0.97 / 1.75 / 1.93 | 1.67 | -9.6 |
| 14 | 2022-03-29 | L1 | 14:16-14:40 | 1.4 | -257 / 288 | 0.74 / 2.00 / 2.32 | 1.21 | -5.2 |
| 15 | 2022-03-29 | L2 | 19:06-19:16 | 0.9 | 53 / 321 | 1.46 / 1.91 / 2.06 | 1.82 | -7.1 |

[a] cloud weighted average



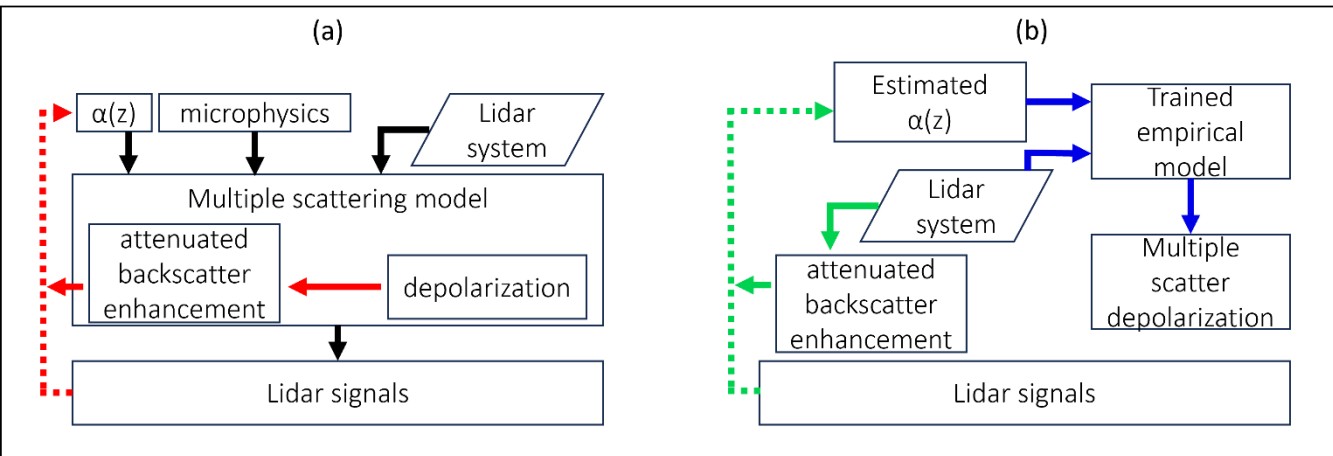

Figure 1: (a) A generic forward model flow diagram for polarization lidar measurements of dense water clouds (black lines) and the use of the depolarization – multiple scattering relationship to retrieve extinction (red lines). (b) System dependent method for estimating the depolarization associated with water clouds as part of a polarization cloud phase retrieval.






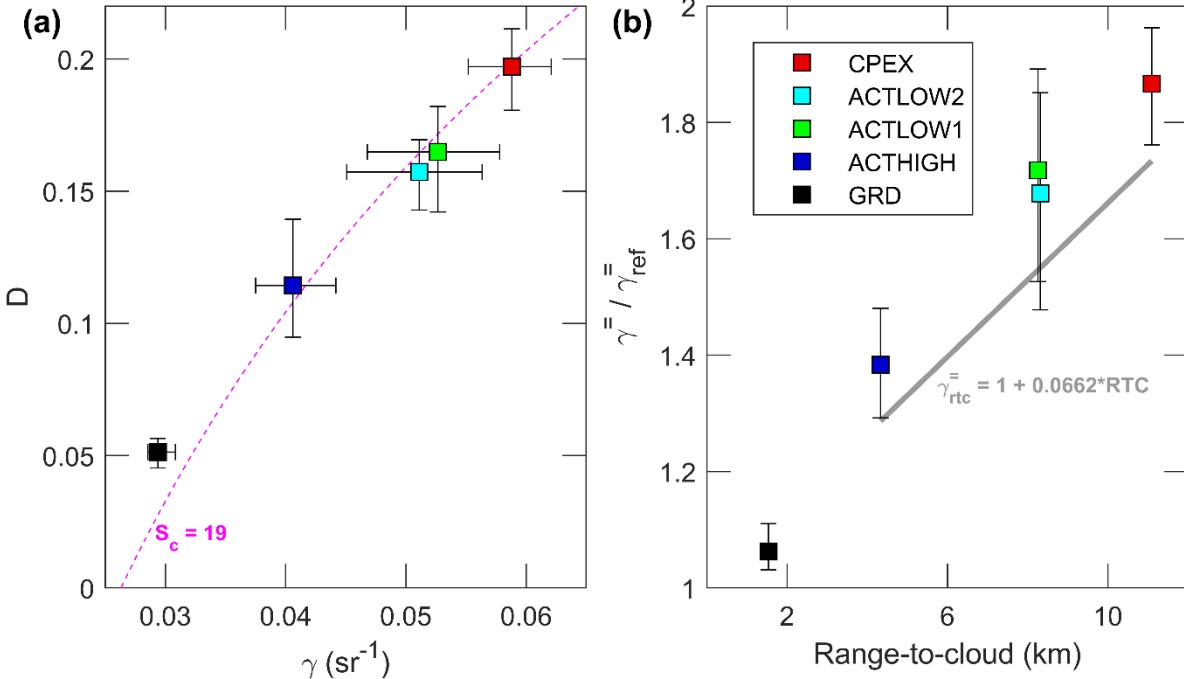

Figure 2: (a) Opaque water cloud layer integrated depolarization and backscatter enhancements caused by multiple scattering showing the relationship with the Hu parameterization at Sc = 19, and (b) the effect of range-to-cloud (RTC) on co-polarized multiple scattering backscatter enhancement. A linear fit though the nadir-facing data points (see text) at the 10[th] percentile is used to establish a minimum multiple scattering threshold.



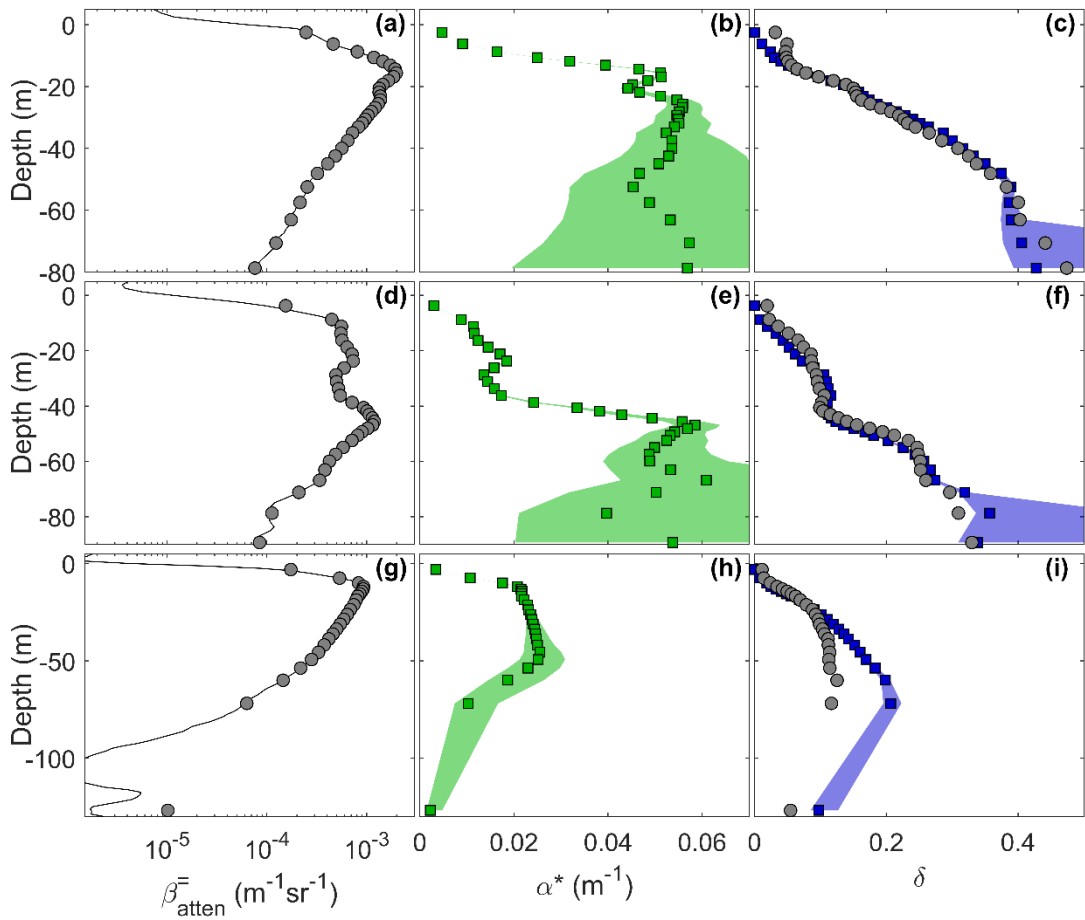

Figure 3: (a-c) Example profiles showing the measured co-polarized attenuated backscatter ($\beta^{\parallel}_{atten}$), estimated extinction ($\alpha^*$)
and compared MSD model with measured depolarization ratio ($\delta$) for a cloud profile with relatively simple structure. (d-f) As
(a-c) except that the example cloud profile contains a marked upward jump in extinction. (g-i) As (a-c) except for an example
translucent cloud profile. In the panels showing the profiles of $\alpha^*$ and $\delta$ the shaded region shows the sensitivity of a $\pm5\%$
change to S* in the retrieval.



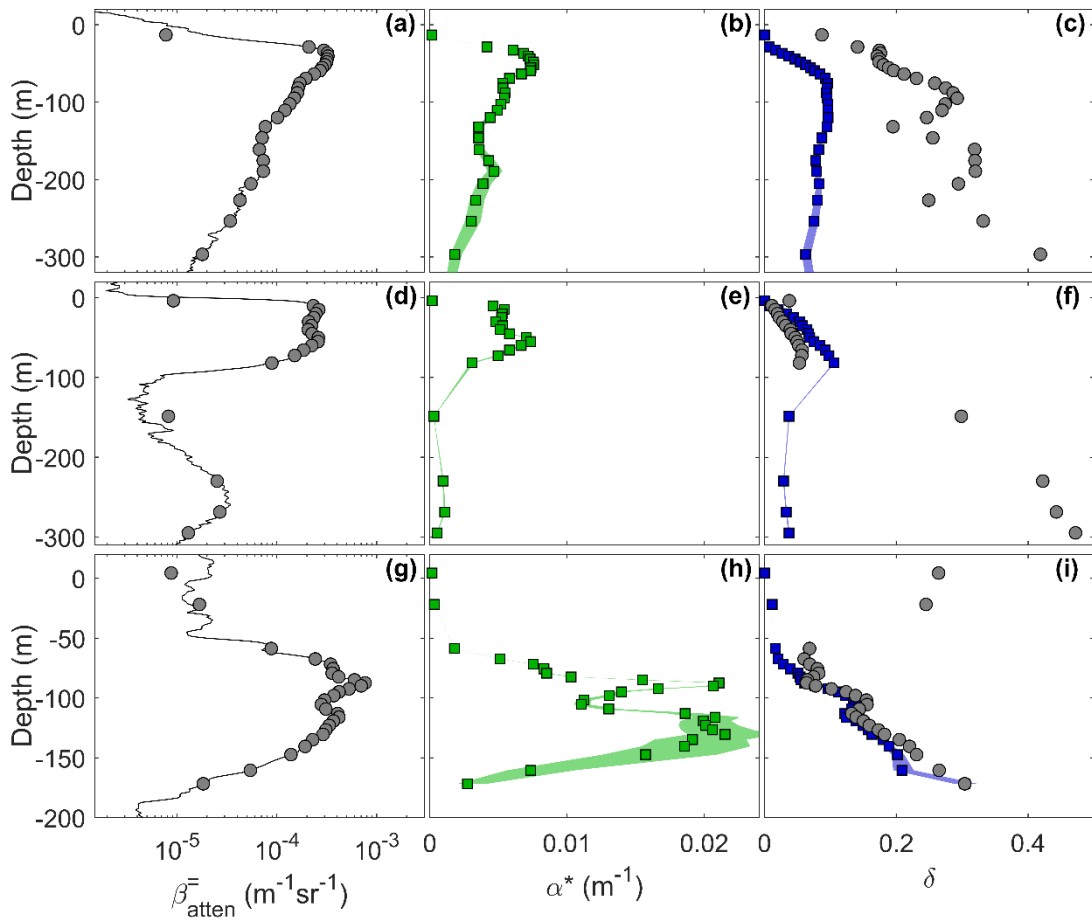

Figure 4: As Figure 3 but for (a-c) a mixed phase profile, (d-f) a profile with liquid-dominant conditions above ice-dominant, (g-i) a profile with ice-dominant conditions above liquid-dominant





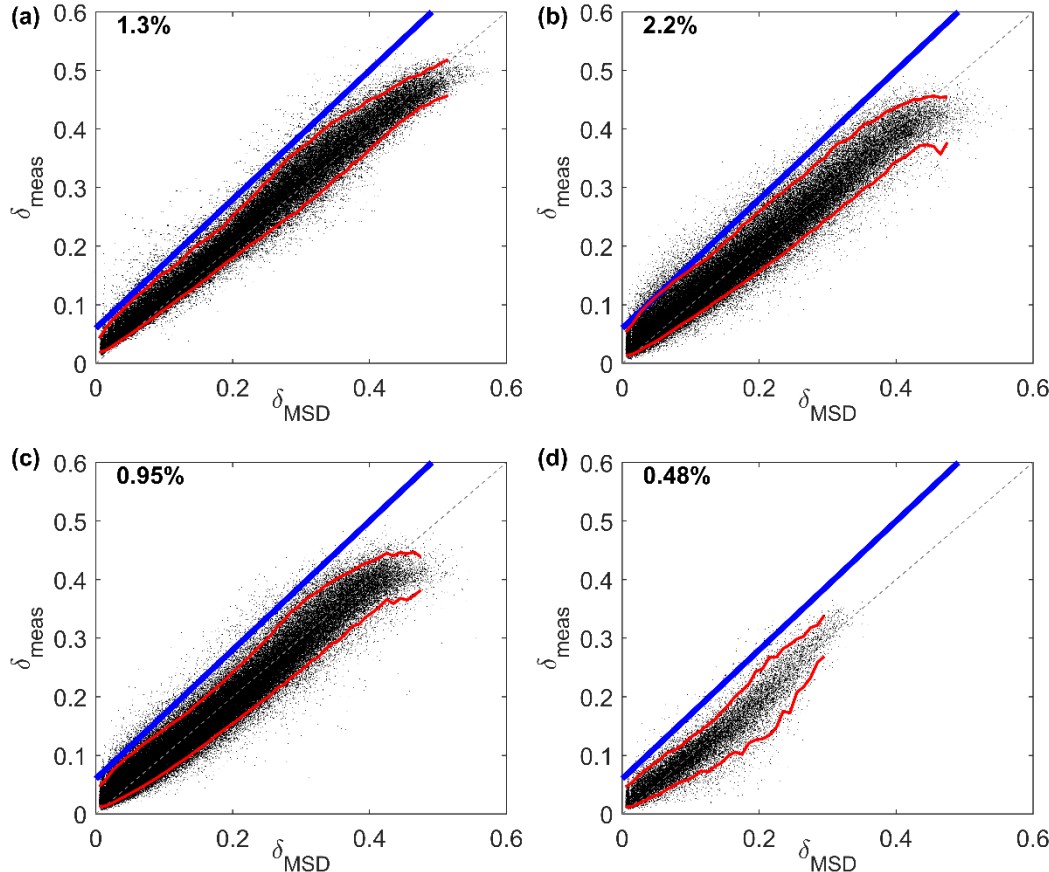

Figure 5: Comparison of the MSD modelled with the measured δ for control cases (a) CPEX, (b) ACTLOW1, (c) ACTLOW2, and (d) ACTHIGH. In each panel a 1:1 line is included (grey dash), 5% and 95% bounds in the measured δ for increments of MSD (red), the boundary of the WATER-MIX classifications (blue), and the false positive fraction in the upper left.





Figure 6: An ACTIVATE flight segment from February 28, 2020, showing the matched King Air and Falcon cloud data. (a) HSRL phase categorization (colors), cloud top height (black), Falcon flight track (grey) and a nearby dropsonde used to provide the temperature scale highlighting the inversion structure (right axis). (b) Number concentrations of microphysical classes: droplets ($< 50\,\mu m$), supercooled large drops (SLD; $> 90\,\mu m$) and ice ($> 90\,\mu m$). (c) Extinction fraction of microphysical classes for periods in cloud. (d) Selected samples of 2D-S particle imagery corresponding to the locations marked in (a).







Figure 7: As Figure 6 but for a flight segment on April 2, 2021.






Figure 8: As Figure 6, but for a flight segment on January 27, 2022. Note that Falcon classes also include the drizzle (DZ) class substituting SLD for altitudes lower than 0˚C.



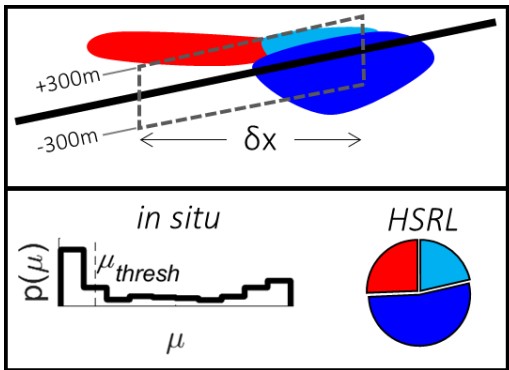


Figure 9: Cartoon vertical cross section diagram showing the matchup method for comparing in situ cloud measurements with the HSRL cloud phase classification. Within each comparison subsegment, denoted by the dashed box, the frequencies of HSRL cloud phase categories (WATER = red, MIX = cyan, ICE = blue) are compared to the distribution of in situ ice extinction fraction.






Figure 10: Comparison of the frequency of (a) ice-containing HSRL categories (MIX, ICE, HOI) and (b) ice-dominant categories (ICE, HOI) with the frequency of Falcon observations that exceed an ice extinction fraction of 0.14 and 0.76, respectively. Each comparison point represents a subsegment (see text) of a matchup cloud scene where the scene indices (Table 3) are shown in the legend. (c) Sensitivity of the ice extinction fraction thresholds, as assessed by the mean absolute

error (MAE) and correlation coefficient (R) of the data shown in (a) and (b). (d) Sensitivity of the subsegment length ($\Delta x$) when the ice extinction fraction thresholds are held at their optimal values.