# Peer review of "A method to retrieve mixed phase cloud vertical structure from airborne lidar"

_EGUsphere, 2024_

## Author Comment (AC1)

**Response to reviewers**

The authors thank both reviewers and provide responses to their comments in blue text below:

**REVIEWER 1**

First review of "A method to retrieve mixed phase cloud vertical structure from airborne lidar" by Crosbie et al.,

January 15, 2025

**Reviewer Recommendation:**

Somewhere between Minor and Major revisions required

**Summary:**

The submitted manuscript introduces an empirical model in an attempt to separate the origin of depolarization observed with a lidar. In my understanding, the authors take observations that are expected to be from liquid only clouds, as screened by temperature, and tune an empirical model to tightly match the depolarization profile observed. This model accounts for depolarization due to multiple scattering. When taking the same model and applying it to clouds that are not necessarily expected to be liquid only, deviations are then attributed to ice. In general, this approach seems reasonable and caveats and assumptions are fairly well documented.

In a world that is often filled with algorithms that over promise and under deliver, this manuscript is, in my opinion, candid with limitations and expectations. I don't often get to write that and I think it is important to note. In general, the paper is well written; it has generally excellent clarity and citations are appropriate in my opinion. It fits well within my understanding of the scope of AMT and I believe it is a contribution that should be published.

I note some elements that I find unclear, and I believe some revisions will improve the draft. That said, it is my recommendation that this manuscript be accepted with revisions (somewhere between major and minor).

**Major Comments:**

1. My prime concern is that I am unclear how far to trust the validation of this technique. In general, I am sympathetic to the fact that you are using airborne measurements and that they are generally infrequent (especially because in this case you require 2 planes in the same patch of sky simultaneously) and you are trying to see into a region where few technique exist. However, in Figure 6 & 7 the flight track of the Falcon and your measurements seem to be worryingly far away for a worryingly large fraction of time. This seems to be the crux of the problem: lidar has poor penetration depth into clouds but you need coincident measurements to validate the technique. My specific comments/questions are:

   The reviewer raises concerns that are unfortunately a central theme of airborne observations, particularly for dynamically evolving clouds. Despite the best efforts of the ACTIVATE team to coordinate the aircraft, some degree of mismatch in sample volume between the two aircraft (whether in 3-D space or in time) was almost always unavoidable.  We have approached the problem in the best way known to us, including additional corrections (such as approximating the contribution for wind drift) that have not been routinely undertaken in other studies involving the ACTIVATE aircraft matchups.

   It should also be added that as it relates to airborne campaigns, ACTIVATE may offer perhaps one of the best datasets as it relates to spatially coordinating two aircraft with over 70% of the nearly combined 1200 flight hours with two aircraft involving them being within 6 km and 5 min of each other; beyond this point we should note that we were careful in this study to use data that involved sufficiently close spatiotemporal coordination to do this analysis robustly. Thus, while not perfect, the spirit of advancing science and technology should use datasets such as this one that are as good as available and meets the standards of the investigating team with decades of experience in this line of work.

   1. It might be best to specify how much total data overlap there is (in terms of hours or kilometers or measurements or something else), i.e. when the Falcon is within the realm of your measurements that you trust (to my novice eye, 15-20 km range on Figure 6 seems to be reasonable overlap but there seems to be almost none from 30-40 km on the same figure).

      We have added an overlap duration metric as an additional column in Table 3 and we thank the reviewer for this suggestion.  For additional clarification, the quantitative evaluation discussed in Section 4 (and shown in Figure 10) would not include almost all of the 30-40 km region that the reviewer mentions, because it does not meet the overlap

requirement (illustrated in Figure 9).  The examples described in Section 3-5-3.7 and shown in Figures 6-8 were picked to highlight a diversity of cloud types.  In the center part of Figure 6, the qualitative agreement in the pattern seen near cloud top with the lidar compared to the deeper in situ is worth stating, but of course we agree that it should not be used as validation. We have made an adjustment to the text in Section 3.5 to clarify this further.

In Section 3.6, (the reviewer mentioned Figure 7 in the introductory remarks for this comment) we have already provided a caveat to any regions that were not vertically collocated but we feel that our commentary/interpretation of the cloud scene is appropriate and adds to a contextual understanding of the overall cloud processes. Again, any regions that do not satisfy the screening criteria are not included in the quantitative validation in Section 4.

2. Lines 428-429: It is clear to me that you are picking +-300 meters as a threshold because you need to have some reasonable sample size. However, it is unclear to me how sensitive to this limit (+- 300 meters) your results are? 300-meter penetration into liquid clouds seems optimistic. Your Figure 3 & 4 suggests that your penetration depth into clouds may reach this threshold but can also be limited to 50-60 meters. As I read this, I think either: 1) your results need to be shown to be robust for various depths of observation or 2) the sensitivity to that change needs to be established. At minimum, if you use a more restrictive threshold, I believe you need to comment on what it does to your interpretation of results.

The $\Delta z$ sensitivity was investigated but was not ultimately included, mainly for brevity. We include the sensitivity analysis below.  As the reviewer notes in the first comment above, it is essential to establish a vertical threshold, but the best choice of value is uncertain.  Smaller $\Delta z$ results in more conservative screening but that does not necessarily translate into a fairer validation because the sample size may become unworkably small, and it amplifies biases related to subjective altitude choices for the Falcon.

While the penetration depth is often less than 300 m (as noted), the local variability in the cloud top is often of a similar magnitude, at least in some of the cloud scenes.  Along a similar argument as made for using a larger $\Delta x$ to minimize residual collocation errors, the larger $\Delta z$ also helps to

suppress collocation error. If the collocation error could be completely eradicated allowing record-by-record comparisons ($\Delta x \sim 50$ m), then one could expect $\Delta z < 10$ m.

Figure 1 shows how the MAE for the mix classification with $\mu_{thresh}=0.14$ changes with $\Delta z$, where $\pm\Delta z$ is the vertical depth threshold. The MAE remains nearly constant through approximately 160 m then drops at lower $\Delta z$. Figure 2 shows how MAE varies with $\mu_{thresh}$ for the mix classification. The behavior is generally quite consistent although the location of minimum varies in the 0.08-0.14 range. For $\Delta z = 200$-500 m, similar MAE performance could be claimed by using $\mu_{thresh}$ anywhere with that 0.08-0.14 range. Figure 3 shows the number of subsegments (defined by $\Delta x$ and $\Delta z$) that are suitable for making comparisons between lidar and in situ. A subsegment needs enough overlapping data points ("pixels") and sufficient cloudy data from the in situ sampling to be considered suitable. As $\Delta z$ drops below 200 m, the number of subsegments drops considerably. The mean number of "pixels" per subsection shown on the right axis of Figure 3 also shows a decrease because the capture area $\Delta x$ by $\Delta z$ decreases.

Our conclusion is that between 200-500 m the choice of $\Delta z$ is insensitive and the result is consistent. Using a smaller $\Delta z$ gives the illusion of better performance (under the metrics for performance we have used) but we believe that it is driven by the precipitous loss of qualifying data points rather than an actual improvement. It is possible that this could be explored further with a future cloud dataset that has more uniform cloud top heights and less concern for collocation error.

We have added the following to the description in L433:

"The analysis was found to be insensitive to vertical zone size in the 200-500 m range."

[Figure]

3. Line 380-382: Given that you are using these examples as validation for your method, it seems like a non sequitur to specify that the method

works and that the Falcon's data can't be used because it was too far away.

This comment is addressed in the responses to 1 and 2 above. To reiterate, segments of the data that do not meet the collocation requirements are screened out from the validation in Section 4. We still feel it appropriate to discuss the time series and show examples like these because it provides context for a reader. We repeatedly caveat any interpretation when there is not direct collocation.

4. I presume the GRD case from Table 1 has no flight validation data. Is that true? If that is true...is it a fair control case? If not, you should more clearly say how that data is compared/validated. For example..where is your temperature data coming from?

None of the water-only control cases introduced in Section 2.2.1 overlap with the King Air – Falcon matchups from ACTIVATE. So, in that sense, none of the control cases are validated with in situ data. The control cases were picked from a range of conditions that crucially spanned as much of the range-to-cloud extent as possible. The temperature was determined from radiosondes or dropsondes.

2. Section 2.3.1 and Lines 412-414: My understanding of the observation of oriented ice crystals is that their properties are highly dependent on the angle of observation. For example looking at Noel and Chepfer 2010, it is my understanding that a change from 0.3 degrees from nadir and 3 degrees from nadir basically removed CALIOP's sensitivity to oriented ice crystals. With that being said, are your observational legs done at nadir exactly or is there some angle of attack? It the plane pitching/rolling in any way that would change your sensitivity? Does that affect your measurements of oriented ice crystals? A comment clarifying your sensitivity would be warranted in my opinion.

Noel, V., and H. Chepfer (2010), A global view of horizontally oriented crystals in ice clouds from Cloud-Aerosol Lidar and Infrared Pathfinder Satellite Observation (CALIPSO), J. Geophys. Res., 115, D00H23, doi:10.1029/2009JD012365

There is some variability across different platforms, but the King Air typically flew with a 2-4 degree nose up pitch angle and the lidar was registered within 0.5 degrees of the fuselage datum. Note, for clarification, it is the nadir offset angle of the lidar relative to vertical that is the critical angle (the angle-of-attack mentioned by the reviewer is the angle of the airframe relative to the airstream and therefore incorporates climb/descent angles).

The maneuvering of the aircraft (pitch/roll) will certainly affect the sensitivity to signatures of oriented ice, as the reviewer correctly notes. However, it is not clear how to provide a statement/comment on this sensitivity, other than to state that the aircraft is oriented off-nadir. We have added this to Section 2.3.1. Even though the lidar is off-nadir (typically at a comparable angle to CALIOP), the ability to identify regions that may be "polluted" by the complicating influence of HOI was still deemed to be important.

As described in L307-308 we did not expect a strong influence from oriented ice in the ACTIVATE matchup cases because most of the mixed phase cloud occurrences were during post-frontal/cold air outbreak conditions. These conditions are turbulent, high LWC environments and therefore promote riming that is assumed to further suppress the HOI signature.

The comments in L412-414 are intended as a cautionary statement and to potentially stimulate further evaluation of the angular dependence of specular reflections. It is possible that with more detailed vertically-resolved information (compared to layer-integrated approaches) the conclusions about the prevalence of oriented ice (and mixing relationships) could evolve. However, we feel that further analysis is out of scope.

**Minor Comments:**

1. Lines 122-124: You mention cross talk in the molecular channel due to imperfect extinction by your HSRL filter. This physical effect is absent your equation 2. I would suggest including that term.

   No, because the series Eq 1-13 explain how we manipulate the signals for this method. We do not use the SR, via Eq (4), for quantitative evaluation of the cloud extensive properties. The only place that SR is used is to establish a reference cloud top altitude. At SR=50 the cross talk is assumed negligible.

2. Equations 10/11/13: It is unclear to use the same variable (z) as both the integrand and the upper bound of integration. Suggest integrating over z' or some other stylized version to differentiate your two variables.

   We added text to clarify that the depth coordinate z starts at cloud top and extends down into the cloud. The addition of other stylized variants on dz was deemed to be more confusing especially for Equation 13 where we already have the superscript ‖. The prime has already been used to denote differentiation with respect to depth in Equations 17 and 18. An asterisk has already been used for Equations 14-16. We feel that the text clarification is sufficient.

Line 192: I would expect this value to be lower (like maybe 8 to 10). I think therefore that a citation for the lidar ratio used here is needed.

References added.  Certainly, by conventional definitions, 18-20 is typical.

3.  Table 1: What is "Cloud Temperature"? Is it really cloud-top temperature or average temperature or something else? Please clarify.

It is an estimate of the temperature where the measurement was made.  Cloud top temperature would be appropriate except for the zenith case (GRD), where we do not necessarily know where cloud top is.  The point of listing these temperatures is to demonstrate that they are not frozen. Given that it is not really the main point, it started to get laborious to further document additional details.

4.  Table 2: I would suggest adding the RMSE value that would be calculated if using your "Final" coefficient values as another row. It would be helpful to see how much your averaged coefficients alter the fit to each test case.

Added, thank you.

5.  Table 2: Why is your GRD case included in Table 1 and not here?

While it was desirable to have an additional data point at close range to confirm the near-field multiple scattering shown in Figure 2, it did not seem appropriate to train the MSD model on zenith data.  While it would be interesting to assess any differences in the accumulation of depolarization from the (typically) more gradual increase in extinction found at cloud base, it was deemed less relevant to the intended end goal.

6.  Figure 2: I would suggest changing your colors for each case to perhaps use different symbols (diamonds, triangles and so forth). For example, I am having a hard time differentiating ACTHIGH and GRD. Additionally, I am not Red-Green colorblind but CPEX and ACTLOW1 might be difficult to differentiate for someone that is.

We have implemented the different symbols suggested by the reviewer and this should circumvent any ambiguity caused by color. Thank you for this suggestion.

7.  Figure 3: Why is there a huge gap in measurements from 60 meters to 120 meters? Is that last data point truly trustworthy or are your data filters kicking in with just one point serendipitously falling into the end?

*From L217-219: "For implementation of the method, Δz is allowed to vary to keep increments in γ∥ constant (up to the native acquisition of the measurement Δz = 1.25 m) and this has the advantage of mitigating the decrease in signal-to-noise ratio with depth in the cloud that affects both α\* and δ."*

The lines in Figure 3a,d,g are the "raw" signal while the dots are the re-gridded values.  This clarification has been added to the figure caption to avoid any ambiguity. Thank you for mentioning it.

8. Figure 3: It might be helpful to add an ordinate axis for optical depth instead of just using physical depth.

   We tried this but it was more clumsy than beneficial.

9. Figure 3 & 4: I would include in the caption what the difference between gray and blue does is (specifically looking at panels C/F/I). I presume it is for observed and modeled depolarization, respectively, but it would be good to clarify.

   Agreed, thank you.

The manuscript presents an empirical methodology for identifying ice and liquid water features within atmospheric clouds. Hydrometeor characterization is a well-documented challenge in cloud research and has been addressed in prior studies. However, this work is the first to develop and test a method specifically designed for an airborne lidar system. The study leverages extensive measurement campaigns, incorporating lidar observations and in situ data, adding significant value to the analysis.

While the manuscript outlines the general approach effectively, certain methodological details require further elaboration. Key geometrical considerations, such as the field-of-view and distance to the cloud, were addressed, and the results support the reliability of the approach. Nevertheless, questions remain regarding the quality of the lidar data, particularly the reported depolarization values, which appear anomalously high.

We would lean on the reviewer to provide more quantitative support for their statement that the depolarization values appear high.  We would also like to reiterate that if the reviewer is used to layer accumulated depolarization ratios for clouds, then the maximum range-resolved depolarization ratios will be higher (typically a factor of ~2).  In addition, if the reviewer is used to very narrow FOV systems at close range then the multiple scattering depolarization will be much smaller in those cases.

We further refer the reviewer to Figure 2a of the manuscript. Here the layer integrated depolarization ratio (D) is compared to the layer integrated backscatter (γ).  At different ranges that modulates the amount of multiple scattering, the relationship conforms to the well-established Hu parameterization of water cloud integrated depolarization ratio, for a typical cloud lidar ratio. If the depolarization ratio were anomalously high, this agreement would not hold.

The integrated depolarization (Eq 13) of the CPEX case shown in Figure 3c is 0.197.  For reference, CALIOP sees layer depolarization sometimes more than 0.3 for water clouds (Hu 2007).

*Hu, Y. (2007). Depolarization ratio–effective lidar ratio relation: Theoretical basis for space lidar cloud phase discrimination. Geophysical research letters, 34(11).*

Additionally, the manuscript suggests that the system is configured to measure only in a downward direction. This setup has significant implications and needs to be explicitly addressed. Unlike ground-based lidars, the downward-facing configuration could lead to depolarization increases due to multiple scattering, which may at some point merge

with the single-scattering depolarization of ice crystals. This is contrary to the typical behavior observed in ground-based systems. Moreover, strong attenuation of the signal in liquid layers near the cloud top might obscure ice or liquid water signatures beneath these layers.

To clarify, these airborne systems are operated in a near-nadir configuration while on the aircraft.  However, they can be operated in a near-zenith configuration in the laboratory. This fact was stated in the manuscript (L183-185) in the description of the ground-based control case.  The manuscript does not suggest that the system is configured to measure only in a downward direction, as the reviewer asserts.

It is challenging to parse the comments in this paragraph from the prior comments that claimed the depolarization was too high for our system. Specifically, it appears that the reviewer is acknowledging that cloud top nadir views may result in more depolarization than they are used to with ground systems. The fact that parts of the scene are obscured because of attenuation is a concern of both upward and downward looking scenarios.

Finally worth noting: yes, absolutely, at some point water multiple scattering depolarization will be equivalent or greater to the typical signature from ice crystals. For some ground-based systems this may never be an issue and therefore could significantly simplify the problem.

To enhance the robustness of the method,  synthetic scenes or ground-based observations might be beneficial in the future.

An excellent suggestion. We hope that data from other systems could be used to train a MSD empirical model using suitable water-only training sets and would welcome comparisons.

**Specific Comments:**

- Line 100: Using the HSRL, backscatter and extinction coefficients can be unambiguously derived. However, how are these parameters affected in liquid clouds? Could crosstalk introduce significant errors in the measurements? Additionally, why was the extinction from the molecular channel not considered in this approach?

  As stated in L122-125, there is a possibility of cross talk at very high SR. This is one of the reasons why the HSRL molecular signal is only used to "calibrate" just above cloud top.  There may be additional information contained within

the molecular signal within the cloud but we are not using it at the current time. This is clearly stated in Section 2.1.5.

- Equations 1, 3, and 5: These equations assume an idealized polarization system. How do the authors ensure the absence of polarizing effects, given the potential inefficiencies in each channel? (Mattis et al. 2009, Freudenthaler 2016). Furthermore, the reported depolarization values for liquid layers appear unusually high.

  See earlier comment regarding the depolarization ratios. The waveplate and channel gain ratios are calibrated at least once per flight and these ratios have been found to be very consistent. See Hair et al. 2008 for details (this is referenced in the manuscript).

- Equations 1 and 2: The use of the same constant for both channels requires justification. Since the detectors and optical paths differ, how is it ensured that the constants are identical? And shouldn't the cross-polarized channel in Equation 3 have a different constant? A polarization calibration might be necessary.

  There are different constants. Their ratios are $G_m$ and $G_{dep}$, respectively. The equations already have this included in the determination of the X terms.

- Equations 8 and 9: These equations describe the attenuated backscatter from the system to range z, not from the cloud base to z. The left-hand equality seems incorrect as written. Shouldn't the left-hand side be divided by ($T\_norm^2 * T\_m^2$)?

  By cloud base, we assume the reviewer means cloud top (in the context of the way that Section 2.1.5 is explained).

  No. The intent is to write an attenuated backscatter with respect to a reference at cloud top, so the left equality describes what we mean it to mean: cloud attenuated total (molecular + particle) backscatter for co-and cross-polarized components.

- Equations 10–13: These equations do not appear essential for introducing integrated quantities. Please consider removing them to streamline the manuscript.

  Respectfully, we disagree. Splitting the y terms (10-11) in this way is a new approach and so we want to define it clearly. The purpose of Eq 12-13 is to

remove any lingering ambiguity about using range-resolved versus layer accumulated depolarization profiles.

- Line 198: The explanation regarding larger lidar ratios for ice crystals underestimating extinction more than liquid clouds is unclear. While multiple scattering contributes to extinction underestimation, this applies to both cases, ice or liquid. Please clarify. What reference supports Equation 16, and what are its limitations? The equation seems to assume a constant extinction coefficient profile, so additional reasoning is required.

  It is not obvious what needs to be clarified in L198. If the lidar ratio is greater than the estimate, the true extinction will be more than the estimate.

  The form of Equation 16 is a modification of the method of Roy and Cao (2010) which is referenced in this section. The limitations are specified throughout Section 2.2.2. It is also made clear that this is an extinction profile and therefore a function of z (not a constant as the reviewer suggests). The lidar ratio is a constant.

- Equation 19: With five constants to retrieve, how good is the performance of this retrieval? Given the potential for multiple solutions, how do the authors ensure an optimal and unique solution?

  For clarification, not all are constants: $r_2$ is a function of range-to-cloud. The quality of the parameters is assessed with respect to the RSME shown in Table 2. The reviewer is justified in asking about the potential for additional suitable parameters. There were correlations amongst the parameters that resulted in zones within the parameter space with similar performance and caused slow convergence. After mapping these dependences, a "brute force" iteration was employed to minimize the risk of a better solution somewhere else.

  Our intention is not to claim that the form of Eq19 is the best representation of multiple scattering. Indeed, there could be physically based or machine learning approaches that are superior. The potential for multiple solutions in the current set of parameters is not overly concerning because we are not trying to interpret their meaning in a physical sense. The uncertainty generated by the RMSE in Table 2 or the bounds in Figure 5, would be good to improve upon but the current levels do not preclude its usage.

- Figure 3: The differences among the three cases are difficult to discern. Extending the vertical range in the figure could enhance interpretability.

The range shown currently spans the entire range of workable data.  In the first (a,b,c) and second (d,e,f) cases the retrieval would not be performed any deeper because we are already at the extinction limit. In the third case (g,h,i), which is the translucent case, we have already exited the cloud and there is nothing more useful.

- Section 3.3: The utility of the retrieved microphysical information is unclear. The primary instrument, the 2DS optical array, only detects hydrometeors larger than 90 μm. This limits its ability to observe cloud droplets, which are typically smaller than this threshold.

  The reviewer has missed some critical aspects in their reading of this section.  First, as listed in the first sentence, both the FCDP and 2DS are used to interpret the microphysics.  The FCDP provides particle distributions from 3-50 μm (stated in L345).  The 2DS can detect particles smaller than 90 μm; however, it cannot distinguish between liquid and ice (based on sphericity) below 90 μm (as stated in L347).  We have no further response to the reviewer's comment.

- Sections 3.5–3.7: When introducing the retrieval scheme, it would be beneficial to first present the lidar-observed variables (e.g., attenuated backscatter, depolarization).

  Perhaps we could include lidar curtains, but there is already quite a large amount of information held in Figures 6-8 and at this stage in the manuscript we wish to focus on the comparisons between the derived phase and the in-situ data.

- Figure 5: Depolarization values up to 50% in warm liquid clouds seem unusually high. Such values are typically associated with ice crystals. What depolarization values are observed for ice crystals?

  Please refer to the earlier response to the alleged high bias in depolarization. The ice threshold was set at 0.35 (L295).

  Therefore, deep into dense clouds it would be ambiguous, if not impossible, to isolate an ice signature.  This is not a fact that we have ignored in the analysis and development of the method.  In the validation section, the conclusion that an ice extinction fraction of 14% defines the onset of the MIX classification is essentially stating that smaller amounts of ice are, on average, not possible to isolate from the signature of multiple scattering from water. With more aircraft matchups we may be able to stratify the data by optical depth (or

some other measure) to improve the interpretation of the threshold. Currently, the datasets are too sparse.

- Finally, the manuscript would benefit from a detailed author contribution statement. Given the technical complexity of the work and the number of coauthors, such a statement would clarify the specific roles of each contributor.

  The author contribution statement is accurate as written.

References:

Mattis, I., Tesche, M., Grein, M., Freudenthaler, V., and Müller, D.: Systematic error of lidar profiles caused by a polarization dependent receiver transmission: quantification and error correction scheme, Appl. Optics, 48, 2742–2751, 2009.

Freudenthaler, V.: About the effects of polarising optics on lidar signals and the Δ90 calibration, Atmos. Meas. Tech., 9, 4181–4255, https://doi.org/10.5194/amt-9-4181-2016, 2016.

---

## Referee Report (RR1)

**Please find my comments in red after the author's responses.**

The manuscript presents an empirical methodology for identifying ice and liquid water features within atmospheric clouds. Hydrometeor characterization is a well-documented challenge in cloud research and has been addressed in prior studies. However, this work is the first to develop and test a method specifically designed for an airborne lidar system. The study leverages extensive measurement campaigns, incorporating lidar observations and in situ data, adding significant value to the analysis.

While the manuscript outlines the general approach effectively, certain methodological details require further elaboration. Key geometrical considerations, such as the field-of-view and distance to the cloud, were addressed, and the results support the reliability of the approach. Nevertheless, questions remain regarding the quality of the lidar data, particularly the reported depolarization values, which appear anomalously high.

We would lean on the reviewer to provide more quantitative support for their statement that the depolarization values appear high. We would also like to reiterate that if the reviewer is used to layer accumulated depolarization ratios for clouds, then the maximum range-resolved depolarization ratios will be higher (typically a factor of ~2). In addition, if the reviewer is used to very narrow FOV systems at close range then the multiple scattering depolarization will be much smaller in those cases.

We further refer the reviewer to Figure 2a of the manuscript. Here the layer integrated depolarization ratio (D) is compared to the layer integrated backscatter (γ). At different ranges that modulates the amount of multiple scattering, the relationship conforms to the well-established Hu parameterization of water cloud integrated depolarization ratio, for a typical cloud lidar ratio. If the depolarization ratio were anomalously high, this agreement would not hold.

The integrated depolarization (Eq 13) of the CPEX case shown in Figure 3c is 0.197. For reference, CALIOP sees layer depolarization sometimes more than 0.3 for water clouds (Hu 2007).

*Hu, Y. (2007). Depolarization ratio–effective lidar ratio relation: Theoretical basis for space lidar cloud phase discrimination. Geophysical research letters, 34(11).*

The reviewer acknowledges that you consider height-resolved values rather than the integrated depolarization.

Depolarization is closely linked to the multiple scattering effect—generally, the greater the contribution of multiple scattering, the stronger the depolarization. The relationship from Hu et al. (2006), as you mention, is a well-established method for relating integrated depolarization to the single scattering fraction in the integrated attenuated backscatter. As you also note in the manuscript, this relationship is largely independent of geometrical features. However, the extent of multiple scattering, and consequently depolarization, does indeed depend on the lidar's geometrical configuration, the distance to the target, and its optical density.

Regarding Figure 2a: You state that the X-axis represents the integrated (attenuated) backscatter, whereas the caption describes it as the backscatter enhancement. Which description is accurate? I assume it refers to the total integrated attenuated backscatter (or integrated parallel attenuated backscatter?).

Hu et al. (2007) found that the (total) integrated backscatter follows this relationship. $\gamma = \gamma_{ss} * \left(\frac{1+D}{1-D}\right)^2$, where $\gamma_{ss} = (1 - T^2)/(2S_c)$. Does the pink line represent this relationship? Please elaborate on this explanation to make it clearer.

Furthermore, the variability range shown in Hu et al. (2007) cover approximately 0.1 - 0.3 for the integrated depolarization $D$ and between 0.05 – 0.1 sr-1 for the attenuated backscatter $\gamma$. Fig 2a in the manuscript shows a range of 0.05 – 0.2 for the integrated depolarization and 0.03 - 0.06 sr-1 for the attenuated backscatter. For an integrated attenuated backscatter of 0.06 Hu et al 2007 reported values of $D$ roughly between 0.13 and 0.17 while you report a mean value of 0.2.

In any case, finding values within a similar range to those of the CALIOP lidar does not necessarily validate the accuracy of the values reported in the manuscript, given the complex nature of multiple scattering in liquid-water clouds. As mentioned, the measured depolarization varies from system to system and depends not only on the cloud's optical thickness but also on factors such as the laser divergence, the receiver's field of view (FOV), the laser and receiver diameters, and the distance to the cloud target. In the case of the CALIOP lidar, multiple scattering effects can play a major role due to its large footprint while orbiting at approximately 705 km altitude.

Modelling studies have reported maximum depolarization values of 0.25–0.35 for a penetration depth of 100–150 meters in relatively opaque clouds (Donovan et al., 2015; Jimenez et al., 2020; Ahmad et al., 2022). However, even these studies consider systems with larger FOVs and small laser divergences (compared to the FOV). When laser divergence and FOV are similar—as appears to be the case for the HSRL-2 and HALO systems—the multiple scattering effect is expected to be weakest (Wang et al., 2021).

In Figure 3c, for instance, depolarization reaches 0.5 at a penetration depth of just 80 meters, which seems surprisingly high.

In my view, whether the depolarization values are overestimated or not does not affect the validity of the proposed methodology. However, I encourage the authors to acknowledge these potential polarization effects. The proposed approach for simulating the depolarization of water clouds (Table 2) would only be applicable to the lidar systems whose data were used for the training.

Hu, Y., Liu, Z., Winker, D., Vaughan, M., Noel, V., Bissonnette, L., Roy, G., and McGill, M.: A simple relation between lidar multiple scattering and depolarization for water clouds, Opt. Lett., 31, 1809–1811, https://doi.org/10.1364/OL.31.001809, 2006.

*Hu, Y., Vaughan, M., Liu, Z., Lin, B., Yang, P., Flittner, D., Hunt, B., Kuehn, R., Huang, J., Wu, D., Rodier, S., Powell, K., Trepte, C., and Winker, D.: The depolarization – attenuated backscatter relation: CALIPSO lidar measurements vs. theory, Opt. Express, 15, 5327–5332, https://doi.org/10.1364/OE.15.005327, 2007.*

*Donovan, D. P., Klein Baltink, H., Henzing, J. S., de Roode, S. R., and Siebesma, A. P.: A depolarisation lidar-based method for the determination of liquid-cloud microphysical properties, Atmos. Meas. Tech., 8, 237–266, https://doi.org/10.5194/amt-8-237-2015, 2015.*

*Jimenez, C., Ansmann, A., Engelmann, R., Donovan, D., Malinka, A., Schmidt, J., Seifert, P., and Wandinger, U.: The dual-field-of-view polarization lidar technique: a new concept in monitoring aerosol effects in liquid-water clouds – theoretical framework, Atmos. Chem. Phys., 20, 15247–15263, https://doi.org/10.5194/acp-20-15247-2020, 2020.*

*Ahmad, W.; Zhang, K.; Tong, Y.; Xiao, D.; Wu, L.; Liu, D. Water Cloud Detection with Circular Polarization Lidar: A Semianalytic Monte Carlo Simulation Approach. Sensors, 22, 1679. https://doi.org/10.3390/s22041679, 2022.*

*Wang, Z., Zhang, J., Gao, H.:Impacts of laser beam divergence on lidar multiple scattering polarization returns from water clouds,*
*Journal of Quantitative Spectroscopy and Radiative Transfer, Volume 268, 107618,*
*https://doi.org/10.1016/j.jqsrt.2021.107618, 2021.*

Additionally, the manuscript suggests that the system is configured to measure only in a downward direction. This setup has significant implications and needs to be explicitly addressed. Unlike ground-based lidars, the downward-facing configuration could lead to depolarization increases due to multiple scattering, which may at some point merge with the single-scattering depolarization of ice crystals. This is contrary to the typical behavior observed in ground-based systems. Moreover, strong attenuation of the signal in liquid layers near the cloud top might obscure ice or liquid water signatures beneath these layers.

To clarify, these airborne systems are operated in a near-nadir configuration while on the aircraft. However, they can be operated in a near-zenith configuration in the laboratory. This fact was stated in the manuscript (L183-185) in the description of the ground-based control case. The manuscript does not suggest that the system is configured to measure only in a downward direction, as the reviewer asserts.

It is challenging to parse the comments in this paragraph from the prior comments that claimed the depolarization was too high for our system. Specifically, it appears that the reviewer is acknowledging that cloud top nadir views may result in more depolarization than they are used to with ground systems. The fact that parts of the scene are obscured because of attenuation is a concern of both upward and downward looking scenarios.

Finally worth noting: yes, absolutely, at some point water multiple scattering depolarization will be equivalent or greater to the typical signature from ice crystals.

For some ground-based systems this may never be an issue and therefore could significantly simplify the problem.

To enhance the robustness of the method, synthetic scenes or ground-based observations might be beneficial in the future.

An excellent suggestion. We hope that data from other systems could be used to train a MSD empirical model using suitable water-only training sets and would welcome comparisons.

Specific Comments:

• Line 100: Using the HSRL, backscatter and extinction coefficients can be unambiguously derived. However, how are these parameters affected in liquid clouds? Could crosstalk introduce significant errors in the measurements? Additionally, why was the extinction from the molecular channel not considered in this approach?

As stated in L122-125, there is a possibility of cross talk at very high SR. This is one of the reasons why the HSRL molecular signal is only used to "calibrate" just above cloud top. There may be additional information contained within the molecular signal within the cloud but we are not using it at the current time. This is clearly stated in Section 2.1.5.

• Equations 1, 3, and 5: These equations assume an idealized polarization system. How do the authors ensure the absence of polarizing effects, given the potential inefficiencies in each channel? (Mattis et al. 2009, Freudenthaler 2016). Furthermore, the reported depolarization values for liquid layers appear unusually high.

See earlier comment regarding the depolarization ratios. The waveplate and channel gain ratios are calibrated at least once per flight and these ratios have been found to be very consistent. See Hair et al. 2008 for details (this is referenced in the manuscript).

The comment refers to potential polarization effects that are unrelated to the gain ratios. These effects may be induced by optical elements within the system, including the beam expander, telescope, lenses, beam splitters, and polarization optics. I have learned that efforts have been made to account for these effects in the case of the HSRL-2 lidar (e.g., Burton et al., 2015).

Burton, S. P., Hair, J. W., Kahnert, M., Ferrare, R. A., Hostetler, C. A., Cook, A. L., Harper, D. B., Berkoff, T. A., Seaman, S. T., Collins, J. E., Fenn, M. A., and Rogers, R. R.: Observations of the spectral dependence of linear particle depolarization ratio of aerosols using NASA Langley airborne High Spectral Resolution Lidar, Atmos. Chem. Phys., 15, 13453–13473, https://doi.org/10.5194/acp-15-13453-2015, 2015.

• Equations 1 and 2: The use of the same constant for both channels requires justification. Since the detectors and optical paths differ, how is it ensured that the

constants are identical? And shouldn't the cross-polarized channel in Equation 3 have a different constant? A polarization calibration might be necessary.

There are different constants. Their ratios are $G_m$ and $G_{dep}$, respectively. The equations already have this included in the determination of the X terms.

Thanks for clarifying this. It was not immediately obvious to me... So, $c$ is the constant for the parallel channel, $G_m c$ for the molecular channel and $G_{dep} c$ for the cross channel.

• Equations 8 and 9: These equations describe the attenuated backscatter from the system to range z, not from the cloud base to z. The left-hand equality seems incorrect as written. Shouldn't the left-hand side be divided by (T_norm^2 * T_m^2)?

By cloud base, we assume the reviewer means cloud top (in the context of the way that Section 2.1.5 is explained).
No. The intent is to write an attenuated backscatter with respect to a reference at cloud top, so the left equality describes what we mean it to mean: cloud attenuated total (molecular + particle) backscatter for co-and cross-polarized components.

When I read cloud-attenuated backscatter, I interpret it as the attenuated backscatter within the cloud, which you refer to as the corrected signal. I suggest emphasizing this in the text by explicitly stating that you define attenuated backscatter as the corrected signal divided by the total transmission from the system to the cloud top (or cloud base if zenith-pointing).

Equations 10–13: These equations do not appear essential for introducing integrated quantities. Please consider removing them to streamline the manuscript.

Respectfully, we disagree. Splitting the γ terms (10-11) in this way is a new approach and so we want to define it clearly. The purpose of Eq 12-13 is to remove any lingering ambiguity about using range-resolved versus layer accumulated depolarization profiles. Ok!

• Line 198: The explanation regarding larger lidar ratios for ice crystals underestimating extinction more than liquid clouds is unclear. While multiple scattering contributes to extinction underestimation, this applies to both cases, ice or liquid. Please clarify. What reference supports Equation 16, and what are its limitations? The equation seems to assume a constant extinction coefficient profile, so additional reasoning is required.

It is not obvious what needs to be clarified in L198. If the lidar ratio is greater than the estimate, the true extinction will be more than the estimate.
The form of Equation 16 is a modification of the method of Roy and Cao (2010) which is referenced in this section. The limitations are specified throughout Section

2.2.2. It is also made clear that this is an extinction profile and therefore a function of z (not a constant as the reviewer suggests). The lidar ratio is a constant.
Yes, of course, the lidar ratio is set as a constant and not the extinction. I understand that. What is not entirely clear to the modification of the method from Roy and Cao (2010) leading to your Equation 16.

The approach from Roy and Cao (2010) allows for the retrieval of the true extinction coefficient by correcting for multiple scattering using the Hu relation. In contrast, your approach accounts for multiple scattering in a more rudimentary manner by using the ratio of integrated attenuated backscatter to the maximum value (or the simulated $\gamma_{\mathrm{rtc}}$). I can understand that you do not necessarily need to perform a sophisticated retrieval of the true extinction coefficient correcting the multiple scattering effect, instead you need $\alpha^*$ for your training and later identification, but equations 14-16 do not look obvious to me.

The term $\alpha^*$ represents the apparent extinction coefficient that produces the measured attenuated signal. To my understanding, the retrieved backscatter (and consequently, the retrieved extinction) will be lower than the true backscatter, regardless of whether the cloud is composed of liquid or ice. This is due to the apparent reduction in attenuation caused by the multiple scattering effect.

In the case of ice clouds, the extinction may be even more underestimated because the backscatter is multiplied by an incorrect lidar ratio to derive extinction from backscatter. This seems logical to me when considering the Klett-Fernald retrieval; however, it is not entirely clear to me in the context of Equations 14–16.

You write in Line 198: *As an example, α\* would underestimate the true extinction profile for conditions where Sc>Sc,ref that may occur with sufficient ice content; however, as will become more apparent, the underestimate usually benefits rather than hinders ice 200 discrimination.*

I may have just missed the part where the benefits become more apparent.

Equation 19: With five constants to retrieve, how good is the performance of this retrieval? Given the potential for multiple solutions, how do the authors ensure an optimal and unique solution?

For clarification, not all are constants: $r_2$ is a function of range-to-cloud. The quality of the parameters is assessed with respect to the RSME shown in Table 2. The reviewer is justified in asking about the potential for additional suitable parameters. There were correlations amongst the parameters that resulted in zones within the parameter space with similar performance and caused slow convergence. After

mapping these dependences, a "brute force" iteration was employed to minimize the risk of a better solution somewhere else.

Our intention is not to claim that the form of Eq19 is the best representation of multiple scattering. Indeed, there could be physically based or machine learning approaches that are superior. The potential for multiple solutions in the current set of parameters is not overly concerning because we are not trying to interpret their meaning in a physical sense. The uncertainty generated by the RMSE in Table 2 or the bounds in Figure 5, would be good to improve upon but the current levels do not preclude its usage.

Figure 3: The differences among the three cases are difficult to discern. Extending the vertical range in the figure could enhance interpretability.

The range shown currently spans the entire range of workable data. In the first (a,b,c) and second (d,e,f) cases the retrieval would not be performed any deeper because we are already at the extinction limit. In the third case (g,h,i), which is the translucent case, we have already exited the cloud and there is nothing more useful.

• Section 3.3: The utility of the retrieved microphysical information is unclear. The primary instrument, the 2DS optical array, only detects hydrometeors larger than 90 μm. This limits its ability to observe cloud droplets, which are typically smaller than this threshold.

The reviewer has missed some critical aspects in their reading of this section. First, as listed in the first sentence, both the FCDP and 2DS are used to interpret the microphysics. The FCDP provides particle distributions from 3-50 μm (stated in L345). The 2DS can detect particles smaller than 90 μm; however, it cannot distinguish between liquid and ice (based on sphericity) below 90 μm (as stated in L347). We have no further response to the reviewer's comment.

I was indeed referring to the ability to distinguish between droplets and ice crystals. The 2DS is unable to identify particles between 50 and 90 μm in diameter. If the FCDP detects particles smaller than 50 μm, you classify the pixel as a droplet, even though some ice crystals may be present.

Given these limitations and the challenges in finding suitable cases, as mentioned in Section 2.4, it was difficult to assess and evaluate the validation approach. Introducing supercooled liquid, drizzle, and droplets at that stage of the manuscript seemed unnecessary. After all, your empirical approach is focused solely on identifying ice and liquid water.

• Sections 3.5–3.7: When introducing the retrieval scheme, it would be beneficial to first present the lidar-observed variables (e.g., attenuated backscatter, depolarization).

Perhaps we could include lidar curtains, but there is already quite a large amount of information held in Figures 6-8 and at this stage in the manuscript we wish to focus on the comparisons between the derived phase and the in-situ data.
•       Figure 5: Depolarization values up to 50% in warm liquid clouds seem unusually high. Such values are typically associated with ice crystals. What depolarization values are observed for ice crystals?

Please refer to the earlier response to the alleged high bias in depolarization. The ice threshold was set at 0.35 (L295).
Therefore, deep into dense clouds it would be ambiguous, if not impossible, to isolate an ice signature. This is not a fact that we have ignored in the analysis and development of the method. In the validation section, the conclusion that an ice extinction fraction of 14% defines the onset of the MIX classification is essentially stating that smaller amounts of ice are, on average, not possible to isolate from the signature of multiple scattering from water. With more aircraft matchups we may be able to stratify the data by optical depth (or some other measure) to improve the interpretation of the threshold. Currently, the datasets are too sparse.

I realize I may not have expressed myself clearly. I was referring to the range of depolarization values you observed in ice crystals. While I understand the difficulty of isolating the ice signature from the multiple scattering effects of water at low ice extinction fractions, what range of values do you typically observe in clear ice crystal cases? This information would also be relevant for the manuscript, alongside the the values for liquid clouds (which are at least provided for the reference pure liquid cases in fig 5).

Finally, the manuscript would benefit from a detailed author contribution statement. Given the technical complexity of the work and the number of coauthors, such a statement would clarify the specific roles of each contributor.

The author contribution statement is accurate as written.
References:
Mattis, I., Tesche, M., Grein, M., Freudenthaler, V., and Müller, D.: Systematic error of lidar profiles caused by a polarization dependent receiver transmission: quantification and error correction scheme, Appl. Optics, 48, 2742–2751, 2009.
Freudenthaler, V.: About the effects of polarising optics on lidar signals and the Δ90 calibration, Atmos. Meas. Tech., 9, 4181–4255, https://doi.org/10.5194/amt-9-4181-2016, 2016.

Line 443 refers to satisfy Equation 22. It should write Equation 21.